# Debiasing Online Preference Learning via Preference Feature Preservation

## Abstract

While various preferred features determine human preferences, current preference learning frameworks for large language models (LLMs) simplify them with binary pairwise comparisons and scalar rewards. This simplification could make LLMs' responses biased to mostly preferred features such as longer responses which would be exacerbated in online learning scenarios as the biases can be accumulate continuously throughout the iterations. To address these challenges, we propose a novel framework called PFP (**P**reference **F**eature **P**reservation). The key idea of PFP is maintaining the distribution of human preference features throughout the online preference learning process. Specifically, PFP first trains a feature classifier using the existing offline pairwise human preference data. Then, using this classifier and the distribution preserving optimization, PFP maps appropriate preference features for each input instruction during online learning. Lastly, PFP trains LLM using the existing preference learning framework, by incorporating the preference feature of each data into system prompts and enabling LLM to explicitly handle various human preferences. Our experiments demonstrate that PFP successfully mitigates the bias in preference features that arise during online learning, and achieves superior performance compared to previous preference learning methods on general benchmarks including AlpacaEval 2.0 and MT-Bench. We also observe that PFP almost resolves a length bias issue, a long-standing problem of online preference learning, even though it was not specifically designed to tackle this.[1]

## 1 Introduction

Aligning large language models (LLMs) using human feedback, particularly by learning from human preferences, yields remarkable successes in various NLP tasks and real-world applications such as coding assistants and chatbots (Anthropic, 2024; Dubey et al., 2024; OpenAI, 2024b; Team et al., 2023). To improve the alignment of LLMs, various preference learning algorithms, such as Reinforcement Learning from Human Feedback (RLHF) (Ouyang et al., 2022) and Direct Preference Optimization (DPO) (Rafailov et al., 2023), have been explored. A common assumption across these works is that human preference is provided in a binary pair-wise comparison (Ziegler et al., 2019; Hong et al., 2024). This approach enables easy modeling of human preference using the scalar reward such as the Bradley-Terry (BT) model (Bradley & Terry, 1952); however, it also has critical limitations from over-simplification and fails to capture the complexity of human preferences.

One typical mis-aligned behavior of LLMs trained under current preference learning methods is a *length bias*; as shown in Fig. 1(a), LLMs tend to produce and prefer longer responses after the alignment procedure (Park et al., 2024; Dubois et al., 2024; Singhal et al., 2023). Despite various attempts to mitigate this issue, such as heuristically penalizing response length within reward models (Chen et al., 2024), length bias remains a persistent and challenging problem. In addition, beyond the bias toward the specific preference feature, another key challenge is a bias toward the preferences of the majority (Santurkar et al., 2023). As the reward model likely assigns higher rewards to the responses preferred by the majority, aligned LLMs with this reward model could be also biased. It makes LLMs suffer to generate the proper responses for diverse users with various preferences.

This issue becomes even more problematic in online preference learning scenarios, which progressively improves the alignment of LLMs by iterating the generation of preference data and learning

---

[1]We will release the codes and models upon acceptance.

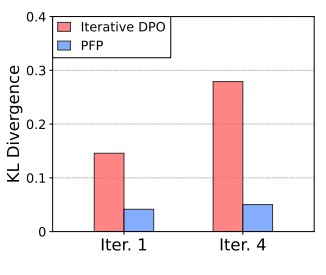 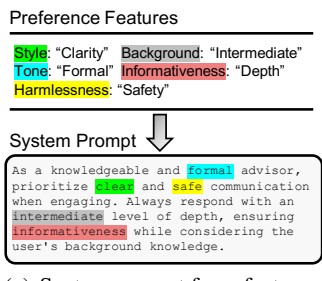

| (a) Length bias | (b) Biased feature distribution | (c) System prompt from features |

Figure 1: **Motivation for debiasing online preference learning.** (a) The average length of the response from LLMs trained with the existing online preference learning is progressively increased with more iterations. (b) Underlying preference feature distribution obtained by inversely asking GPT-4o is progressively biased toward the majority at the initial distribution. Larger KL divergence indicates that the feature distribution has shifted further from its pre-training state. (c) We propose to map each input instruction with the specific preference features and then convert it into the system prompt to enable LLM to explicitly handle preference features. (See detail in Sec. 5.1)

from them (Xiong et al., 2024; Wu et al., 2024; Rosset et al., 2024). During online preference learning, LLM will generate responses biased toward specific preference features, and the preference annotators, such as the external reward model (Jiang et al., 2023b), will provide positive feedback on this. As such iterations go on, the bias of LLM accumulates (see Fig. 1(b)), and hence it results in the reduced diversity and quality of LLM's responses.

**Contribution.** To address these challenges, we propose a novel online preference learning framework called PFP (**P**reference **F**eature **P**reservation). Our approach is to ensure that the distribution of preference features remains consistent throughout the online preference learning process. Here, the key idea is to explicitly extract preference features of each input instruction and handle them using system prompts of LLMs (see Fig. 1(c)); it enables LLMs to generate and learn preference data with intent. Specifically, PFP first estimates the initial distribution of preference features of the given human preference dataset, by inferring which features mainly determine binary human preferences. We then train a preference feature classifier, which maps each input instruction to appropriate preference features with additional optimization for the distribution preservation, during the online learning process. Finally, PFP trains LLM using the existing preference learning framework, by converting the mapped preference features of each generated data into the system prompts of LLMs.

We demonstrate the effectiveness of the proposed PFP by applying it to align recent open-sourced LLMs, *e.g.*, Mistral (Jiang et al., 2023a), with the commonly used preference dataset (UltraFeedback Cui et al. (2023)) and evaluation benchmarks (AlpacaEval 2.0 (Dubois et al., 2024) and MT-bench (Zheng et al., 2023)). For the experiments, we adopt the SELFEE framework (Kim et al., 2024) as an online preference learning algorithm, which enables Iterative DPO (Xu et al., 2023; Xiong et al., 2024) without using an external reward model. Our experimental results demonstrate that PFP successfully eliminates the bias in preference features during online learning. As shown in Fig. 1, responses generated by the model trained with PFP mitigate bias in preference features, unlike the model trained with Iterative DPO. Additionally, PFP achieves superior performance compared to previous online preference learning methods. For example, our framework achieves 7.58% → 15.24% increase in AlpacaEval 2.0 length-controlled win rate compared to the SFT model, while Iterative DPO achieves 7.58% → 13.13% increase. More interestingly, PFP effectively reduces the occurrence of length bias during online preference learning, despite not being specifically designed to address this issue. These results demonstrate that our framework is highly competitive and practical for real-world applications, underscoring the robustness and versatility of the proposed framework.

## 2 RELATED WORKS

**LLM alignment with human preference.** Aligning LLMs with human intentions and values using human feedback data now becomes a defacto standard to obtain well-performing LLMs (Ziegler et al., 2019; Ouyang et al., 2022). Typically, this feedback is collected by asking human annotators to compare two responses generated from the same input prompt and assign a binary preference label based on their judgment. One of the most widely adopted approaches is RLHF (Christiano

et al., 2017; Stiennon et al., 2020), where a reward model is trained to capture human preferences (Bradley & Terry, 1952), and the LLM is then fine-tuned to optimize for this learned reward. To prevent issues such as reward over-optimization and model collapse, KL divergence regularization is commonly employed during this fine-tuning process. However, RLHF presents several challenges such as computational overheads from training reward models, as well as the instability associated with online reinforcement learning algorithms. To address these issues, alternative approaches have been extensively proposed (Rafailov et al., 2023; Zhao et al., 2023; Meng et al., 2024; Hong et al., 2024); for instance, Rafailov et al. (2023) propose DPO, which eliminates the need for a separate reward model by deriving a training objective that is mathematically equivalent to RLHF.

**Online preference learning.** Existing preference learning methods can generally be categorized into two approaches depending on whether they use the fixed human preference dataset (*offline preference learning*, *e.g.*, DPO) or progressively enlarge dataset from the iterations of sampling and labeling (*online preference learning*, *e.g.*, RLHF). While online methods typically achieve superior performance due to train with more data, they also demand significantly more computational costs from sampling responses and labeling preferences. To address this challenge, recent work has focused on developing efficient batch-online preference learning techniques, such as Iterative DPO (Xu et al., 2023; Xiong et al., 2024; Rosset et al., 2024; Wu et al., 2024; Calandriello et al., 2024). Iterative DPO generates thousands of responses in each iteration (batch) and constructs labeled preference datasets by judging the preference using the reward model (Jiang et al., 2023b). This dataset is then used to train LLMs with offline methods like DPO, and the iteration repeats, resulting in more efficient and stable alignment.

**Bias of LLMs after alignment.** One prominent issue observed in LLMs after alignment with existing preference learning methods (RLHF and DPO) and binary preference labels is the emergence of a *length bias*, where LLMs tend to generate and favor the longer responses (Park et al., 2024; Singhal et al., 2023). Not only for the trained LLM policy, automated evaluation methods, including reward models and LLM-as-a-judge frameworks, also often exhibit a bias toward longer outputs, complicating the accurate assessment of LLM performance (Dubois et al., 2024; Wang et al., 2023). Although various strategies have been proposed to mitigate length bias, such as incorporating length penalties into the reward function (Park et al., 2024) or adjusting the objective function (Chen et al., 2024), the issue remains difficult to fully resolve. Another key challenge is a bias toward the preferences of the majority (Santurkar et al., 2023) which can yield other unexpected and hidden biases, as the reward model will likely assign higher rewards to the responses preferred by the majority. This issue becomes more problematic in the online preference learning setup, as the bias of LLMs accumulates with more iterations. In this paper, we propose a new approach to mitigate this problem by explicitly extracting the preference features and handling them via system prompt.

## 3 PRELIMINARY: ONLINE PREFERENCE LEARNING

Let the LLM policy be denoted as $\pi_\theta$, which can generate output sequence (*i.e.* response) $y$, given input sequences composed of *system prompt* $s$ and *instruction* $x$, *i.e.*, $y \sim \pi_\theta(s, x)$. Here, the system prompt $s$ is usually considered to be fixed regardless of the input instruction $x$. For convenience, we assume that $s$ is always included as the input of $\pi_\theta$ and hence omit $s$ in the equations in the below parts. Next, we assume that we have the labeled preference dataset, $\mathcal{D} = \{(x, y_l, y_w)\}$, where $y_l$ and $y_w$ are the dis-preferred and preferred responses for the corresponding instruction $x$, respectively.

**RLHF and DPO.** To train $\pi_\theta$ with $\mathcal{D}$ for the alignment, RLHF first introduces the reward model $r(x, y)$ which can convert human preference data into scalar values. Specifically, the reward model $r(x, y)$ is often modeled with the Bradley-Terry (BT) model (Bradley & Terry, 1952), and then it can yield the probability $p(y_w \succ y_l \mid x)$ that response $y_w$ is preferred over $y_l$ as follow:

$$p(y_w \succ y_l \mid x) = \frac{\exp\left(r(x, y_w)\right)}{\exp\left(r(x, y_w)\right) + \exp\left(r(x, y_l)\right)}. \tag{1}$$

As the optimal reward function $r(x, y)$ is not accessible, a parameterized reward model $r_\phi(x, y)$ is usually introduced by optimizing its parameters with the maximum-likelihood objective on the preference dataset. With this reward model, RLHF optimizes LLM $\pi$ to maximize this reward with the additional regularization of the KL divergence between the current policy and the reference policies ($\pi_{\text{ref}}$) to prevent reward over-optimization:

$$\mathcal{L}_{\text{RLHF}}(\pi_\theta, \pi_{\text{ref}}) = -\mathbb{E}_{y \sim \pi_\theta, x \sim \rho}\left[r_\phi(x, y)\right] + \beta \mathrm{D}_{\text{KL}}\left(\pi_\theta(y|x) \parallel \pi_{\text{ref}}(y|x)\right). \tag{2}$$

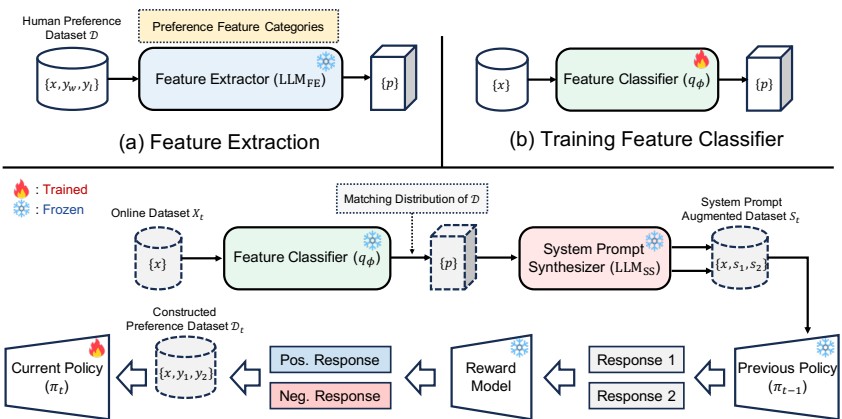

Figure 2: **Illustration of the proposed PFP framework.** (a) PFP first extracts the preference feature among the predefined categories for a given human preference dataset using an LLM-based feature extractor. (b) With the extracted features, PFP trains the feature classifier. (c) The trained feature classifier along with the additional adjustment maps the corresponding preference feature for a new instruction in a given online dataset. Then, the LLM-based system prompt synthesizer converts it into two system prompts, where each system prompt is used to sample the separate response. Then, the labeled preference dataset is constructed and the current policy LLM is trained on this dataset.

To remove the necessity of the reward model in RLHF, DPO proposed a method that is mathematically equivalent to the original RLHF objective and can directly optimize the internal reward modeled by LLM $\pi$ itself, by maximizing the weighted likelihood gap between $y_w$ and $y_l$:

$$p_\theta(y_w \succ y_l|x) = \sigma\left(\beta \log \frac{\pi_\theta(y_w|x)}{\pi_{\text{ref}}(y_w|x)} - \beta \log \frac{\pi_\theta(y_l|x)}{\pi_{\text{ref}}(y_l|x)}\right). \quad (3)$$

$$\mathcal{L}_{\text{DPO}}(\pi_\theta, \pi_{\text{ref}}, \mathcal{D}) = \mathbb{E}_{(x,y_w,y_l)\sim\mathcal{D}}\left[-\log p_\theta(y_w \succ y_l|x)\right]. \quad (4)$$

**Online preference learning and SELFEE.** In the online preference learning scenario, we first assume that we have multiple unlabeled instruction datasets $X_t = \{x\}$, $t = 1,...,T$ where $X_i \cap X_j = \emptyset$ for all $j = 0,...,i-1$. For $t$-th iteration, the preference dataset $\mathcal{D}_t = \{(x, y_l, y_w)|x \in X_t\}$ is constructed by (1) sampling two responses for each instruction $x \in X_t$ using LLM policy $\pi_{t-1}$ from the previous iteration (*i.e.*, $y_1, y_2 \sim \pi_{t-1}(x)$), and (2) judging the preference between them. Then, LLM policy $\pi_t$ which is initialized with $\pi_{t-1}$ is trained with $\mathcal{D}_t$ using the existing preference learning method. One representative approach is Iterative DPO (Xu et al., 2023), where the external reward model is used for the preference judgments and $\pi_t$ is trained with $\mathcal{D}_t$ using DPO.

However, as choosing the proper reward model is non-trivial, especially in our framework, we adopt SELFEE (Kim et al., 2024) as the online preference learning algorithm. Specifically, SELFEE conducts preference labeling using the implicit reward derived from the DPO's objective function, unlike the other Iterative DPO methods using the external reward model:

$$p_{t-1}(y_1 \succ y_2|x) = \sigma\left(\beta \log \frac{\pi_{t-1}(y_1|x)}{\pi_{\text{init}}(y_1|x)} - \beta \log \frac{\pi_{t-1}(y_2|x)}{\pi_{\text{init}}(y_2|x)}\right), \quad (5)$$

$$(y_w, y_l) = (y_1, y_2) \text{ if } p_{t-1}(y_1 \succ y_2|x) > 0.5 \text{ else } (y_w, y_l) = (y_2, y_1), \quad (6)$$

where $y_1$ and $y_2$ as the generated response from $\pi_{t-1}$. With this preference judgment, SELFEE constructs the labeled dataset $\mathcal{D}_t = \{(x, y_l, y_w)|x \in X_t\}$ and uses it to learn $t$-th policy $\pi_t$. In this work, we assume that $\pi_0$ is trained with DPO on the initial human preference data $\mathcal{D}$.[2]

# 4 PFP: DEBIASED ALIGNMENT VIA PREFERENCE FEATURE PRESERVATION

**Overview.** In this section, we present PFP: **P**reference **F**eature **P**reservation to align LLMs by reducing the bias during online preference learning. Our main idea is to explicitly extract preference

---

[2]Following the conventional setup, we initialize this LLM with SFT.

features of input instruction, and handle them using system prompts of LLMs. To this end, PFP first extracts the preference features of the given human-labeled preference dataset (Sec. 4.1). Then, we train the feature classifier using these extracted features; it enables us to map the proper preference feature for the input instruction of the online dataset while preserving the original feature distribution (Sec. 4.2). Lastly, we train LLMs with the extracted features by incorporating them into the system prompt (Sec. 4.3). We present full procedure of PFP in Algorithm 1 (see Fig. 2 for the overview).

## 4.1 EXTRACTING PREFERENCE FEATURE FROM BINARY HUMAN PREFERENCE DATA

We first assume that some features affect the judgment of human preference between the responses for the given input prompt; we call them *preference features*. Following Lee et al. (2024), we predefined these preference features and organized them into five different classes (*e.g.*, tone, style, informativeness, etc.), denoted as $\mathcal{P} = [C_1, C_2, C_3, C_4, C_5]$, as shown in Table 6. Each class $C_k$ contains up to five sub-features, represented as $C_k \in \{c_k^1, c_k^2, c_k^3, c_k^4, c_k^5\}$; for example, in *style*, one of the classes, consists of following five sub-features: clarity, conciseness, format, vividness, and consistency. Under this definition,, we extract the preference features of the pairwise offline human preference data $\mathcal{D}$ using the feature extractor. We implement the feature extractor by prompting LLM such as GPT-4o (OpenAI, 2024b), to infer the likely preference features that led the annotators to provide specific feedback. Specifically, for the input instruction $x$ and the two responses $y_w$ and $y_l$, the feature extractor is defined as $\mathbf{p} = \text{LLM}_{\text{FE}}(x, y_l, y_w)$ where $\mathbf{p} = [p_1, ..., p_5]$, where each $p_i$ represents a probability distribution over the 5 sub-features of class $C_i$ (*i.e.*, $p_i \in [0, 1]^5$ and $\sum_{j=1}^{5} p_i^j = 1$). The prompts used for the feature extraction are detailed in Appendix C. Then, the extracted preference features are added to the human preference data $\mathcal{D}$ and it yields $\mathcal{D}_{\text{FE}} = \{(p, x, y_l, y_w)\}$.

## 4.2 DISTRIBUTION PRESERVED MAPPING OF INPUT INSTRUCTION TO PREFERENCE FEATURE

To preserve the feature distribution over each iteration of online preference learning, we first map each instruction $x \in X_t$ used in online learning to the proper preference features. One can expect that the preference feature distribution is preserved by explicitly utilizing the assigned features during response generation and preference judgment. Specifically, this process involves two key components: (a) learning a feature classifier, and (b) assigning a pseudo-label using a relabeling technique.

**Learning feature classifier.** PFP introduces an auxiliary classifier $q_\phi$ to predict appropriate preference features for the given input instruction. Specifically, $q_\phi$ is trained via conventional supervised learning with cross-entropy loss, using the input instructions $x$ and the extracted features $p$ in $\mathcal{D}_{\text{FE}}$. After the training, $q_\phi$ can provide a probability distribution over preference features for a new input instruction $x \in X_t$ that will be used in online learning. A separate classifier $q_{\phi^k}$ is introduced for each feature class $C_k$, *i.e.*, $q_{\phi^k}(\cdot) : x \to q_{\phi^k}(x)$ where $q_{\phi^k}(x) = [0, 1]^5$ and $\sum q_{\phi^k}(x) = 1$.[3]

**Adjusted output prediction.** To further complement the classifier's predictions to be aligned with the distribution of human preferences, PFP adjusts the predicted probabilities by introducing the optimization problem. Formally, for each feature class $C_k$, the human preference feature distribution is derived from $\mathcal{D}_{\text{FE}}$, *i.e.*, $P_k = \sum_{p \in \mathcal{D}_{\text{FE}}} p_k / |\mathcal{D}_{\text{FE}}|$. Next, the output probabilities for all input instructions in $X_t$ under $q_{\phi^k}$ is collected to measure the distribution, *i.e.*, $Q_k = \sum_{x \in X_t} q_{\phi^k}(x) / |X_t|$. Here, our goal is to find the adjusted output probability $\widetilde{q}_k(x) \in [0, 1]^5$ for each input instruction $x \in X_t$ that yields the empirical distribution identical with $P_k$ while minimizing the deviation from the original probability $q_{\phi^k}(x)$. This problem can be formulated as below optimization problem:

$$\min_q \text{CE}(q_{\phi^k}, q) \quad \text{s.t.} \quad \forall x \in X_t : q(x) \in [0, 1]^5, \sum_{i=1}^{5} q(x)_i = 1, \text{ and } \sum_{x \in X_t} q(x)/|X_t| = P_k \quad (7)$$

where $\text{CE}(q_{\phi^k}, q)$ is a cross-entropy between $q_{\phi^k}(x)$ and $q(x)$ for $x \in X_t$. Following the previous works (Asano et al., 2020; Kim et al., 2020), we solve this problem via efficient Sinkhorn-Knopp algorithm (Cuturi, 2013). With $\widetilde{q}_k(x)$ from solving Eq. 7 with $q_{\phi^k}$, we sample the preference feature and augment the online dataset $X_t$, *i.e.*, $p_k \sim \widetilde{q}_k(x)$ and $\widetilde{X}_t = \{(p, x) | x \in X_t, p = [p_1, ..., p_5]\}$.

---

[3] $q_{\phi^k}$ is initialized with a relatively small language model (304M), DeBERTa-v3-large (He et al., 2023).

---

**Algorithm 1** PFP algorithm

---

**Input:** initial LLM $\pi_{\text{init}}$, human preference dataset $\mathcal{D}$, number of online learning iterations $T$, new instruction sets $\{X_t\}_{t=1}^T$, feature extractor $\text{LLM}_{\text{FE}}$, system prompt synthesizer $\text{LLM}_{\text{SS}}$

---

Extract preference features of $\mathcal{D}$ using $\text{LLM}_{\text{FE}}$ and construct $\mathcal{D}_{\text{FE}}$ (Sec. 4.1)
Training feature classifier $q_\phi$ using $\mathcal{D}_{\text{FE}}$ (Sec. 4.2)
$\pi_0 \leftarrow \text{DPO}(\pi_{\text{init}}, \pi_{\text{init}}, \mathcal{D}_{\text{FE}})$ through Eq. 4
**for** $t = 1$ **to** $T$ **do**
    Assign preference features for $x \in X_t$ using $q_\phi$ and solving Eq. 7, and construct $\widetilde{X}_t$
    Sample two system prompts $s_1, s_2$ for $p \in \widetilde{X}_t$ using $\text{LLM}_{\text{SS}}$, and construct $S_t$
    Synthesize preference data $\mathcal{D}_t$ with $\pi_{t-1}$ and $S_t$ (Eq. 5 and 6)
    $\pi_t \leftarrow \text{DPO}(\pi_{t-1}, \pi_{t-1}, \mathcal{D}_t)$ through Eq. 4
**end for**
**return** $\pi_T$

---

### 4.3 LEARNING PREFERENCE FEATURES THROUGH SYSTEM PROMPT

**Synthesizing system prompt from preference feature.** We need to generate responses and judge the preference using the LLM policy $\pi_\theta$ conditioned on the given preference feature. However, it can be difficult as the preference features have the form of short words that are not suitable for LLM, for example, the feature set is represented as follows: [Conciseness, Formal, Accuracy, Intermediate, Efficiency]. To address this, we convert these discretized preference features into the system prompt, which is a natural language description about the preference feature, and add it in front of the instruct $x$ as the usual system prompt (see Sec. 3). Specifically, the system prompt $s$ is created through the system prompt synthesizer, which is realized by prompting LLM that receives features as input and generates a system prompt, *i.e.*, $s \sim \text{LLM}_{\text{SS}}(p)$. Then, we augment the online learning dataset $\widetilde{X}_t$ by incorporating the generated system prompt, *i.e.*, $S_t = \{(s, x) | x, p \in \widetilde{X}_t\}$. We created the prompt for $\text{LLM}_{\text{SS}}$ by modifying the prompt used in Lee et al. (2024) (see Appendix C). Using $S_t$, one can perform the existing online preference learning method, such as iterative DPO.

**Double system prompt sampling and scheduling.** While incorporating preference features into LLM using the system prompt enables LLM to understand and handle them better, we observe that conditioning specific system prompts could reduce the diversity between sampled responses. This reduced diversity makes preference judgment between them difficult and consequently leads to decreased performance (see Table 3). To prevent this, we propose to augment the online learning data set $X_t$ by sampling two system prompts, *i.e.*, $S_t = \{(s_1, s_2, x) | x \in X_t\}$ and $s_1, s_2 \sim \text{LLM}_{\text{SS}}(p)$. Then, during the dataset construction process, each system prompt is used to sample the different response, *i.e.*, $y_i \sim \pi_{t-1}(s_i, x)$ where $i = 1, 2$. Finally, using Eq. 5 and 6, we judge the preference between $y_1$ and $y_2$ with randomly chosen $s$ between $s_1$ and $s_2$, and construct the labeled dataset $\mathcal{D}_t = \{(s, x, y_l, y_w) | x \in X_t\}$ for $t$-th iteration which is used to learn the $t$-th policy $\pi_t$.

In addition, to improve the effectiveness of online preference learning, we propose progressively increasing the training examples' difficulty. To this end, we simply reduce the temperature used for system prompt sampling as the iteration increases, which reduces the diversity between two system prompts. We expect that it also reduces the distance between two responses $y_1$ and $y_2$ from online response sampling with $\pi_{t-1}$ and $S_t$, *i.e.*, more difficult to learn; therefore, this approach improves the effectiveness of online preference learning by continuously increasing the difficulty of the task.

## 5 EXPERIMENTS

### 5.1 EXPERIMENTAL SETUPS

In this section, we first present our experimental setups. As denoted in Sec. 3, we adopt the SELFEE framework (Kim et al., 2024) as our online preference learning algorithm for the experiments. SELFEE enables the effective alignment of LLMs with limited preference data and does not require the external reward model; SELFEE includes the process of using initial seed data to train and create

the initial DPO model. Here, the initial DPO model acts as the base model as well as the reward model before the iterative learning process begins (Eq. 5 and 6).

**Models.** For the policy LLM, we utilize an open-source model fine-tuned (SFT) on UltraChat data (Ding et al., 2023) based on the Mistral-7B-0.1v model (Jiang et al., 2023a), following the Zephyr recipe.[4] For the feature classifier (Sec. 4.2), we employ DeBERTa-v3-large (He et al., 2023) as the backbone. We create five separate classifiers, one for each class of preference feature.

**Datasets.** For the initial labeled preference data, we use UltraFeedback dataset (Cui et al., 2023) which has been extensively used by prior works (Snorkel, 2024; Rosset et al., 2024; Kim et al., 2024). Specifically, we sample 10K samples to construct a seed dataset. For PFP, the seed data would be taken feature extraction and system prompt synthesis processes, and the resulting data with added system prompts are used for initial DPO training and feature classifier training. Excluding seed data, we sample 4 datasets of 5K data samples each, ensuring no overlap. These datasets contain only prompts and are used to generate responses in each iteration of online learning.

**Baselines.** To evaluate the effectiveness of PFP, we consider *DPO* (Rafailov et al., 2023), *Iterative DPO* (Xiong et al., 2024), and *SELFEE* (Kim et al., 2024) as the baselines. All models under different baselines are trained starting from the same SFT model. The DPO trains LLM on the seed data without mapped system prompts. Iterative DPO, SELFEE, and PFP used the same online instruction datasets for each iteration. For the reward model (RM) in Iterative DPO, we employed the PairRM (Jiang et al., 2023b), which is wildly used in alignment task. While the initial DPO model was originally adopted as a base model only for PFP and SELFEE, we also consider using initial DPO as a base model in the case of Iterative DPO for a fair comparison. Specifically, we train the initial DPO model using seed data (without mapped system prompts) for SELFEE and Iterative DPO.

**Evaluations.** To measure the basic performance of the model, we employ commonly used benchmarks in preference alignment research, AlpacaEval 2.0 (Dubois et al., 2024) and MT-Bench (Zheng et al., 2023). AlpacaEval 2.0 is designed to approximately evaluate human preference for instruction following, and calculates the win rate by comparing the response of GPT-4 (OpenAI, 2023) and the target model response by using GPT-4 as the evaluator. It is known that this benchmark reflects human preferences well, including a length-controlled win rate that reduces the impact of length bias. On the other hand, MT-Bench is designed to evaluate more diverse capabilities of LLMs by utilizing GPT-4 to score the responses of the model under evaluation on a scale from 0 to 10. In addition, to measure the debiasing effect on preference features, we extract the preference features from the responses generated for the test instructions in AlpacaEval 2.0. Then, we use the GPT-4o (OpenAI, 2024a) to infer the most prominent preference feature in each response. After obtaining the feature distribution, we measure how the KL divergence between this distribution and the feature distribution of the initial model's responses. Here, the initial model refers to the model before the online iteration.

$$D_{\text{KL}}(P_{\text{Initial Model}} \parallel P_{\text{target}}) = \sum_x P_{\text{Initial DPO}}(x) \log \left( \frac{P_{\text{Initial Model}}(x)}{P_{\text{target}}(x)} \right) \tag{8}$$

**Implementation details.** We extract preference features using the GPT-4o on the seed data. Here, the temperature is set to 0 to employ zero-shot chain-of-thought (CoT) reasoning (Wei et al., 2022; Kojima et al., 2022). The feature classifiers are trained to predict the labels of preference features extracted from the seed data, taking the instructions as input (*i.e.*, sequence classification). The number of labels is set to 5, corresponding to the number of sub-features. We train the classifiers with a learning rate of 1e-5, a batch size of 32, over 5 epochs. We synthesize system prompts also using the GPT-4o, taking preference features as input. For double prompt sampling and scheduling (Sec. 4.3), the system prompts in the first iteration are generated with a temperature of 1.25, decreasing by 0.25 with each subsequent iteration. If scheduling is not applied, system prompts are generated with a temperature of 1. For subsequent iterations and the initial DPO, we set $\beta = 0.1$ and train for 1 epoch with 32 batch size. This value is the same throughout PFP and SELFEE learning, but in the case of Iterative DPO, $\beta = 0.01$ was used during online learning. The learning rate of 5e-7 is used with AdamW optimizer (Loshchilov et al., 2017). We employ a cosine learning rate scheduler with a 0.1 warm-up ratio of total running step. For PFP, Iterative DPO, and SELFEE, response sampling was performed twice per prompt with a temperature of 0.7. Unlike the original SELFEE, we removed the self-refine step to reduce the number of tunable hyper-parameters and ensure the robustness of the experiments. The prompt which used GPT-4o is provided in Appendix C.

---

[4] `alignment-handbook/zephyr-7b-sft-full`

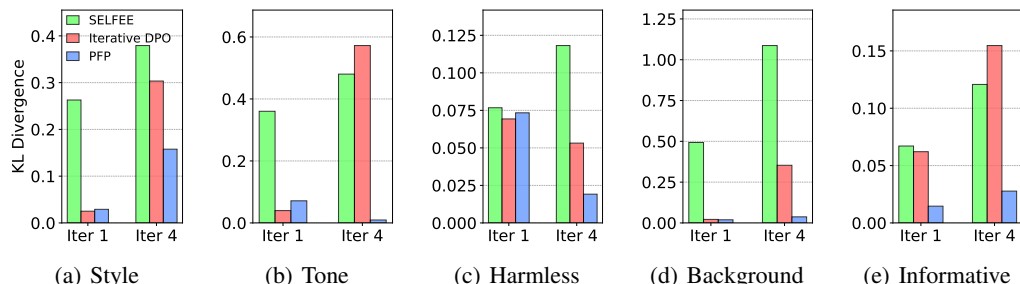

(a) Style  (b) Tone  (c) Harmless  (d) Background  (e) Informative

Figure 4: **Change of preference features.** KL divergence comparison by class, showing how the feature distribution of the initial DPO model's response evolves during the online learning process. PFP, unlike other iterative learning algorithms, shows minimal change in distribution.

Table 1: **Main results.** Evaluation results on AlpacaEval 2.0 and MT-Bench with different variants of Mistral-7B-v0.1. The best scores are highlighted in **bold**.

| Models | AlpacaEval 2.0 | | | MT-Bench |
|---|---|---|---|---|
| | Len-control. Win Rate (%) | Win Rate vs. GPT-4 (%) | Avg. len (# chars) | Avg. Score (0-10) |
| Mistral-7B-v0.1 | 0.17 | 0.50 | 5692 | 3.25 |
| SFT | 7.58 | 4.72 | 901 | 6.34 |
| DPO (W/o sys) | 9.93 | 8.02 | 1409 | 6.34 |
| DPO (W sys) | 9.27 | 5.86 | 1135 | 6.61 |
| SELFEE | 14.23 | **17.49** | 2412 | 6.56 |
| Iterative DPO | 13.13 | 12.05 | 1709 | 6.53 |
| PFP (Ours) | **15.24** | 10.18 | **1187** | **6.88** |

## 5.2 MAIN RESULTS

We compare a DPO model trained with the preference feature from human feedback data explicitly included in the system prompt, against a model trained without feature. Based on AlpacaEval 2.0, the model trained with the system prompt performs slightly worse (9.93 vs 9.27), but based on MT-bench, a model trained with the system prompt gets a higher score than others (6.34 vs 6.61) (see Table 1). However, as shown in Fig. 3, which KL divergence is measured from the SFT response distribution, the DPO model with the preference feature exhibits significantly reduced preference feature bias, and the length bias is also considerably decreased. Specifically, compared to the SFT model's response length of 901, the model trained without system prompts yields an average response length increase to 1409, while the model trained with system prompts only increases to 1135. These results suggest that explicitly considering the preference feature from human feedback data into the system prompt significantly aids in debiasing the model.

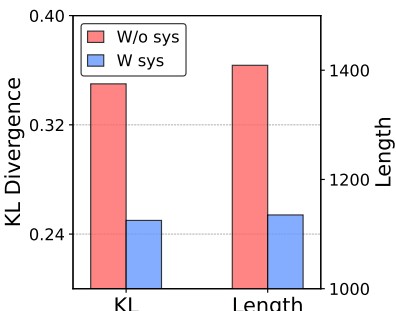

Figure 3: **Initial DPO Analysis.** LLMs trained by DPO using human feedback data with system prompt has less length and feature distribution bias.

We also compare PFP to Iterative DPO and SELFEE; we measure the performance of the model obtained after performing a total of 4 iterations presented in Table 1. PFP succeeded in achieving higher performance than SELFEE (7.58 → 14.23), Iterative DPO (7.58 → 13.13) with a performance improvement of (7.58 → 15.24) based on AlpacaEval 2.0 length-controlled win rate. In MT-Bench, PFP also showed a large improvement (6.34 → 6.88) compared to SELFEE, with (6.34 → 6.56) and Iterative DPO (6.34 → 6.53). This shows that PFP learning achieves performance that surpasses SELFEE or Iterative DPO even on common benchmarks such as AlpacaEval2.0 or MT-Bench.

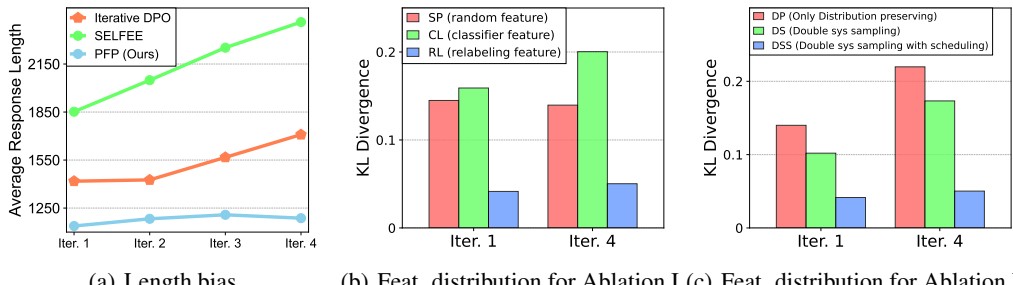

(a) Length bias     (b) Feat. distribution for Ablation I   (c) Feat. distribution for Ablation II

Figure 5: **Analyses.** (a) Length bias with different methods, (b,c) feature distribution for ablation.

Table 2: **Ablation study for feature classifier and distribution preserving.** Evaluation results on AlpacaEval 2.0 and MT-Bench with iteratively trained models (from initial DPO) under different methodological configurations of PFP. SP, CL, RL are abbreviations of system prompt, classifier label, and relabeling, respectively. When using only the system prompt, features are mapped randomly.

| Methods | Method | | | AlpacaEval 2.0 | | | MT-Bench |
| --- | --- | --- | --- | --- | --- | --- | --- |
| | SP | CL | RL | Len-control Win Rate (%) | Win Rate vs. GPT-4 (%) | Avg. len (# chars) | Avg. Score (0-10) |
| PFP (Ours) | ✓ | ✗ | ✗ | 12.38 | 8.99 | 1129 | 6.84 |
| | ✓ | ✓ | ✗ | 14.80 | 10.57 | 1277 | 6.76 |
| | ✓ | ✓ | ✓ | 15.24 | 10.18 | 1187 | 6.88 |

Fig. 4 further describes the changes in the preference feature distribution of responses throughout the iteration process which are measured with KL divergence through Eq. 8. In the case of Iterative DPO and SELFEE, the distribution continues to change, while in the case of PFP, the marginal change in distribution occurs as iteration progresses. This represents that the existing iterative improvement algorithm has bias at the feature level, and PFP sufficiently alleviates this.

Fig. 5(a) describes the changes in the response character length throughout the iteration process. From iteration 1 to iteration 4, the response length for Iterative DPO and SELFEE increased significantly ($1418 \rightarrow 1709$) and ($1852 \rightarrow 2412$), respectively. In contrast, PFP exhibited only a minimal increase in length ($1138 \rightarrow 1187$). This highlights that, unlike other iterative improvement algorithms that have a weakness at length bias, PFP learns human preferences well without causing length bias.

### 5.3 ABLATION STUDY I: FEATURE CLASSIFIER AND DISTRIBUTION PRESERVING

To evaluate the effect of the feature labeling method, we removed some of the feature labeling methods and conducted an ablation study. Table 2 shows the experimental results of performance changes according to differences in feature labeling methods. The results are measured after a total of 4 iterations. Here, the random feature is created by generating a preference feature regardless of the prompt, and the classifier feature is sampled based on the probability of the feature generated when receiving the prompt as input using a preference feature classifier. Additionally, we conduct the relabeling of the probability of the features according to Eq. 7 to preserve the distribution. As a result of the experiment, the feature sample method through the classifier achieves a performance increase of ($12.38 \rightarrow 14.8$) based on AlpacaEval 2.0 compared to the random sampling method, however, based on MT-bench, decreased slightly ($6.84 \rightarrow 6.74$). In the case of the re-labeling algorithm, compared to before re-labeling is applied, a performance increase of ($14.8 \rightarrow 15.24$) is achieved based on AlpacaEval2.0, and a performance increase of ($6.76 \rightarrow 6.88$) is achieved based on MT-bench. Meanwhile, as shown in Fig. 5(b), the bias appears in the preference feature when using the classifier feature. However, when applying the re-labeling algorithm, preference feature bias can be significantly reduced while performance increases.

### 5.4 ABLATION STUDY II: DOUBLE SYSTEM PROMPT SAMPLING AND SCHEDULING

To evaluate the effect of the response sampling method, we conduct experiments by adding double system prompt sampling and scheduling elements. As shown in Table 3, the double system prompt

Table 3: **Ablation study for different system prompt sampling methods.** Evaluation results on AlpacaEval 2.0 and MT-Bench with iteratively trained models (from initial DPO) under different methodological configurations of PFP. DP, DS, DSS are abbreviations of distribution preserving, double system prompt sampling, and double system prompt sampling with scheduling, respectively.

| Methods | Method | | | AlpacaEval 2.0 | | | MT-Bench |
| | DP | DS | DSS | Len-control Win Rate (%) | Win Rate vs. GPT-4 (%) | Avg. len (# chars) | Avg. Score (0-10) |
|---|---|---|---|---|---|---|---|
| PFP (Ours) | ✓ | ✗ | ✗ | 12.73 | 10.10 | 1316 | 6.56 |
| | ✓ | ✓ | ✗ | 13.78 | 9.65 | 1187 | 6.77 |
| | ✓ | ✓ | ✓ | 15.24 | 10.18 | 1187 | 6.88 |

Table 4: **Comparison with baselines to mitigate length bias.** Evaluation results on AlpacaEval 2.0 and MT-Bench with iteratively trained models (from initial DPO) under different methods to mitigate length bias (length penalty and R-DPO). The best scores are highlighted in **bold**.

| Methods | AlpacaEval 2.0 | | | MT-Bench |
| | Len-control Win Rate (%) | Win Rate vs. GPT-4 (%) | Avg. len (# chars) | Avg. Score (0-10) |
|---|---|---|---|---|
| Iterative R-DPO (iter 4) | 13.07 | 11.36 | 1613 | 6.80 |
| Iterative DPO (iter 4) | 13.13 | **12.05** | 1709 | 6.53 |
| PFP (Ours) | **15.24** | 10.18 | **1187** | **6.88** |

sampling yields a large performance improvement, with AlpacaEval 2.0 (12.73 → 13.78) and MT-Bench (6.56 → 6.77). Not only the performance improvement, but the response length also decreased (1316 → 1187 tokens). When scheduling is further applied, the improvement is enlarged, with AlpacaEval 2.0 (12.73 → 15.24) and MT-Bench (6.56 → 6.88). Additionally, these components not only improve performance but also play a significant role in bias mitigation. As shown in Fig. 5(c), double system prompt sampling and scheduling greatly reduce feature distribution bias. In terms of length bias, compared to the case without these components, the additional component reduces the response length (1316 → 1187). These results demonstrate that double system prompt sampling and scheduling are key factors that both enhance performance and mitigate bias.

## 5.5 LENGTH BIAS

The way PFP reduces length bias is fundamentally different from the traditional length control methods. In general, length bias has been handled using heuristic methods. The lengt penalty method works by heuristically subtracting a bias based on the length in the reward term from the reward model (Dong et al., 2024). Alternatively, as seen in the R-DPO approach (Park et al., 2024), length bias can be mitigated by the adding length regularization into the DPO loss. The common point is that the difference in length between two sentences is simply processed heuristically. However, we point out that the method tends to be sensitive to hyper-parameters and often fails to work effectively in practice. To evaluate how well PFP manages length control, we compare PFP with the length penalty method and R-DPO applied to Iterative DPO. We have tried both methods, and found that the R-DPO method with $\alpha = 0.01$ was best applied method. Details about the experiment are in Appendix B. As shown in Table 4, the overall reduction in length remained limited. This shows that PFP is more effective in controlling length compared to traditional methods.

## 6 CONCLUSION

In this paper, we propose PFP, a novel framework that explicitly preserves preference features during the online preference learning process to reduce bias. We demonstrate that incorporating preference features from human feedback into system prompts and preserving the feature distribution over each iteration of online learning effective in preventing bias. This not only aligns human preferences better than the existing Iterative DPO method but also succeeds in almost eliminating length bias and preference features that occur in the learning process. These findings are further supported by various benchmarks and additional analyses.

## REPRODUCIBILITY STATEMENT

For the reproducibility of our results, we have provided a detailed description of our methods and experimental setups in Section 5.1 and Appendix B. In addition, to further facilitate the reproduction, we will release our codes and the checkpoints for the trained models.

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

Table 5: **Results of several length control algorithms and hyperparameter search.** Evaluation results on AlpacaEval 2.0 and MT-Bench with iteratively trained models (from initial DPO) under different methods to mitigate length bias (length penalty and R-DPO). Due to the limited computational budget, we selectively evaluate the models on MT-Bench. The best scores are highlighted in **bold**.

| Methods | AlpacaEval 2.0 | | | MT-Bench |
| | LC Win Rate | Win Rate vs. GPT-4 | Avg. len (# chars) | Score |
|---|---|---|---|---|
| Initial DPO | 9.93 | 8.02 | 1409 | 6.34 |
| Iterative DPO (iter 1) | 10.48 | 8.35 | 1418 | - |
| Iterative DPO (iter 1) w length penalty ($\gamma = 0.01$) | 11.02 | 8.63 | 1433 | - |
| Iterative DPO (iter 1) w length penalty ($\gamma = 0.001$) | 9.60 | 7.72 | 1406 | - |
| Iterative DPO (iter 1) w length penalty ($\gamma = 0.0001$) | 10.72 | 8.55 | 1414 | - |
| Iterative R-DPO ($\gamma = 0.1$) | 9.99 | 8.49 | 1519 | - |
| Iterative R-DPO ($\gamma = 0.01$) | 11.09 | 8.53 | 1385 | - |
| Iterative DPO (iter 4) | 13.13 | 12.05 | 1709 | 6.53 |
| Iterative DPO w length penalty (iter 4) | 12.19 | 11.08 | 1689 | 6.60 |
| Iterative R-DPO (iter 4) | 13.07 | 11.36 | 1613 | 6.80 |
| PɴP (Ours) | **15.24** | 10.18 | **1187** | **6.88** |

## A   LIMITATION AND FUTURE WORK

Extracting preference features and generating system prompts currently requires powerful LLMs like GPT-4o (OpenAI, 2024a), which requires additional computational costs. Future work should explore the use of smaller LLMs such as LLaMA-3-8B (Dubey et al., 2024) for this process. Additionally, further research is needed to assess the impact of incorporating system prompts into the supervised fine-tuning (SFT) stage of training.

## B   BASELINES TO REDUCE LENGTH BIAS DURING ALIGNMENT

**Length penalty.** We applied the length penalty according to the RLHFlow approach (Dong et al., 2024). This is a method to apply a length penalty at the labeling stage by adjusting the reward of the reward model according to Eq. 9. To find the efficient hyper-parameter for this baseline, we experimented with $\alpha = 0.01, 0.001,$ and $0.0001$ for iteration 1. Then, we applied the hyper-parameter that most effectively reduced length ($\alpha = 0.001$, see 3rd-5th rows in Table 5) through iteration 4. As shown in Table 5, this approach often fails. Although $\alpha = 0.001$ showed the best reduction in length in iteration 1, the overall reduction in length remained limited and the performance was degraded as a result. This was the same even when iteration was extended.

$$r_{\texttt{penalty}}(x, y) = r(x, y) - \alpha|y| \tag{9}$$

**R-DPO.** For conduct R-DPO (Park et al., 2024), we change DPO objective function to following Eq. 10. Similar to the length penalty method, we experimented with $\alpha = 0.1, 0.01$ for iteration 1, to find the effective hyper-parameter $\alpha$. As observed in Table 5, $\alpha = 0.01$ successfully reduces the responses' length ($1709 \rightarrow 1613$), but the reduction is still limited to resolve the length bias. These results show that heuristic length control is often unstable and does not work effectively.

$$\mathcal{L}_{\texttt{R-DPO}}(\pi_\theta) = -\mathbb{E}_{(x, y_w, y_l) \sim \mathcal{D}} \left[ \sigma \left( \beta \log \frac{\pi_\theta(y_w|x)}{\pi_{\texttt{ref}}(y_w|x)} - \beta \log \frac{\pi_\theta(y_l|x)}{\pi_{\texttt{ref}}(y_l|x)} \right) + \alpha(|y_w| - |y_l|) \right] \tag{10}$$

## C   PRE-DEFINED PREFERENCE FEATURE SET

Table 6 shows the pre-defined preference feature set $\mathcal{P}$. The definition of the preference feature set was referenced from Janus Lee et al. (2024). Preference features consist of 5 different classes (i.e.

Table 6: **Predefined preference feature set.**

| Domain | Feature Set |
|---|---|
| **Style** | Clarity, Conciseness, Format, Vividness, Consistency |
| **Tone** | Formal, Authoritative, Sophisticated, Engaging, Familiar |
| **Harmlessness** | Sensitivity, Safety, Accuracy, Morality, Trustworthiness |
| **User's Background Knowledge** | Basic, Novice, Intermediate, Advanced, Expert |
| **Informativeness** | Relevance, Practicality, Depth, Creativity, Efficiency |

---

**prompt**

---

Read the following two responses to the same prompt. After reading, determine why the preferred response is chosen over the dispreferred response, focusing on the aspect of {domain}.

**Prompt:** [{prompt}]

**Preferred Response:** [{chosen}]

**Dispreferred Response:** [{rejected}]

### Question
An arbitrary person labeled the responses as preferred and dispreferred.
Considering the aspect of {domain}, what {domain} element does this person likely prefer?

Select one of the following options:

{option}

Finally you have to answer as following format:
-Answer is

Let's think step by step

---

Figure 6: **Prompt for feature extraction.** Input prompt for the feature extraction form pairwise preference data.

Style, Tone, etc), and each class gets 5 different sub-features (i.e. Clarity, Conciseness, etc). Each preference is defined by a total of five sub-features, with one sub-feature assigned per class.

## D  PROMPTS FOR EXPERIMENTS

For the experiments, we construct prompts by modifying the ones used in Lee et al. (2024):

**Feature extraction from human preference data.** Fig. 6 shows the prompts used for extracting preference features from human feedback data. For each class, the prompt is customized to extract a single sub-feature. Only for extracting preference features about the user's background Knowledge, we utilize a differently customized prompt (7).

**Feature extraction from LLM's responses.** Fig. 7 displays the prompts used to measure the preference feature distribution of the responses from LLM. For each class, the prompt is customized to extract a single sub-feature. Similar to the human cases, we utilize a differently customized prompt (9) for the user's background knowledge class.

**System prompt generation.** Fig. 10 shows the prompt used to generate the system prompt based on the input preference feature set. This prompt takes sub-features corresponding to the five classes as input to generate the system prompt.

---

**prompt**

---

Read the following two responses to the same prompt. After reading, determine why the preferred response is chosen over the dispreferred response, focusing on the aspect of the user's background knowledge.

**Prompt:** [{prompt}]

**Preferred Response:** [{chosen}]

**Dispreferred Response:** [{rejected}]

### Question
An arbitrary person labeled the responses as preferred and dispreferred.
What level of background knowledge does the user have that makes them prefer the preferred response over the dispreferred response?

Select one of the following options:

{option}

Finally you have to answer as following format:
- Answer is

Let's think step by step.

---

Figure 7: **Prompt for feature extraction.** Input prompt for the feature extraction from pairwise preference data, focusing on user's background knowledge.

---

**prompt**

---

Given a prompt and a response, analyze the response and determine which preference feature the response was likely based on. Focus on the aspect of {domain}.

**Prompt:** [{prompt}]

**Response:** [{response}]

### Question
An arbitrary person selected this response based on a preference for certain features within the domain of {domain}.
Considering the aspect of {domain}, what specific feature within this domain is the person likely prioritizing?

Select one of the following options:

{options}

Finally, provide your answer in the following format:
- Answer is [selected option Alphabet]

Let's think step-by-step.

---

Figure 8: **Prompt for feature extraction.** Input prompt for the feature extraction form single response of LLM.

# E  QUALITATIVE EXAMPLES

**System prompt sampling.** Fig. 11 illustrates how the preference feature is sampled into the system prompt, using examples from the actual double system prompt process.

---

**prompt**

---

Given a prompt and a response, analyze the response and determine which preference feature the response was likely based on, considering the user's background knowledge.

**Prompt:** [{prompt}]

**Response:** [{response}]

### Question
An arbitrary person selected this response based on a preference for certain features related to their background knowledge. Considering the aspect of the user's background knowledge, what specific feature is the person likely prioritizing?

Select one of the following options:

{options}

Finally, provide your answer in the following format:
- Answer is [selected option Alphabet]

Let's think step-by-step.

---

Figure 9: **Prompt for feature extraction.** Input prompt for the feature extraction form single response of LLM, focusing on user's background knowledge.

Table 7: **Feature classifier accuracy.** Test accuracy of the trained feature classifier (in Section 4.2) on the separately constructed test dataset.

| Metric | background | harmlessness | informativeness | style | tone |
|---|---|---|---|---|---|
| Accuracy | 0.535 | 0.512 | 0.688 | 0.496 | 0.507 |
| F1 Score | 0.532 | 0.513 | 0.663 | 0.489 | 0.426 |

**Examples of generated responses.** Here, we present a direct comparison between Iterative DPO with PFP using the generated responses on the AlpacaEval 2.0 Benchmark. The results are presented in Figures 12, 13, and 14. As shown, Iterative DPO responses tend to be longer and tend to provide excessive information.

## F  ADDITIONAL ANALYSES

**Accuracy of trained preference feature classifier.** In Table 7, we additionally measure the test accuracy of the trained feature classifier on the separately constructed test dataset. Specifically, this test dataset is created by randomly selecting 917 samples from the initial dataset to be excluded from learning. The results show moderate accuracy, which is limited by the small amount of training data and the long-tailed nature of preference features (see Figure 16). These findings highlight the importance of the proposed distribution preservation step to address the feature classifier's limitations.

**Evaluation on Arena Hard benchmark.** To provide a stronger assessment of our framework by evaluating our framework on more diverse datasets, we have additionally evaluated our framework and key baseline methods on the "Arena-Hard-Auto" benchmark (Li et al., 2024), which is another well-established benchmark to evaluate LLM alignment (Meng et al., 2024). The results are presented in Table 8; here, one can observe that while PFP performed slightly below SELFEE, it still outperforms the iterative DPO model by a significant margin. Notably, PFP consistently produced shorter responses compared to both SELFEE and iterative DPO, indicating effective bias mitigation in response length.

**Comparison with SimPO.** Here, we additionally include the comparison with the latest offline preference alignment algorithm, SimPO (Meng et al., 2024), as our new baseline. This baseline is trained using the same seed data size as PFP, which is also the same amount of data used by DPO.

---

**system prompt**

---

You are an excellent system message generator. Read the provided rule, system message examples, and preferences carefully

---

**prompt**

---

I'm brainstorming system messages for personalizing language models. You are given some preferences made by human. 5 preferences are given, and each preference consists of the name of the preference and a description for it. Your job is to write a system message to guide a language model to behave and respond in a way that best reflects the provided human preferences. Please generate a creative and realistic system message. Refer to the given system message examples.

**[Rule]**
- Do NOT include any greeting messages.
- No bullet point style.
- The length of the system message should not be too long. Generate a system message that is about one paragraph in length.
- Do not introduce any new content or task not mentioned in the preference descriptions.
- Do not stick to expressions like "language model", "LLM", "Assistant", and "AI" unless the preference descriptions specifically refer to language model and assistant-related content.
- The system message should assign a role tailored to the preferences to the model.

**[Preferences]**
Style: {style}
Ensure your responses reflect the given style.

Tone: {tone}
Make sure the tone of your responses matches the given tone.

Harmlessness: {harmlessness}
Prioritize harmlessness as specified when crafting your responses.

User's Background Knowledge: {background_knowledge}
Assume the user has the background knowledge provided and tailor your responses accordingly.

Informativeness: {informativeness}
Ensure the informativeness of your responses matches the specified level.

**[Generated system message]**

---

Figure 10: **Prompt for system prompt generation.** Input prompt for the system prompt generation from preference feature.

---

**Preference Features**
'Format', 'Formal', 'Accuracy', 'Basic', 'Depth'

---

**System prompt 1** You are a formal and precise expert, dedicated to providing accurate and harmless information while maintaining clarity and a well-structured format. Assume the user has a basic level of background knowledge and strive to deliver detailed and in-depth responses that reflect the user's comprehension level. Your role is to ensure your responses are meticulously crafted to be both informative and non-harmful, adhering to a consistently formal tone and format

---

**System prompt 2** You are a knowledgeable and formal advisor catering to users with basic background knowledge. Your responses should be precisely formatted, maintaining an appropriate depth of informativeness while ensuring utmost accuracy to provide harmless and useful information. Aim to deliver clear, well-structured, and respectful guidance that thoroughly addresses the queries presented

---

Figure 11: **Result of double system prompt sampling.** Two system prompts are sampled differently from the same preference features.

---

**Prompt:**
What type of soil is suitable for cactus?

---

**Iterative DPO response:**
Cacti are adapted to grow in arid and semi-arid regions with little rainfall, and therefore require well-draining soil that is low in organic matter. A suitable soil for cacti is a well-draining, porous mixture that allows water to drain away quickly and prevents waterlogging.

Here are some characteristics of suitable soil for cacti:

1. Texture: The soil should be coarse-textured with large particles like sand and gravel, as they help with drainage. Cacti do not like their roots to be waterlogged, so a soil mixture with a lot of fine particles like clay can lead to water retention and root rot.

2. pH: Cacti prefer slightly acidic to neutral soil, with a pH range of 6.0 to 7.5. You can test your soil pH using a soil test kit or send a sample to a laboratory for analysis.

3. Nutrient Content: Cacti are not heavy feeders and do not require high nutrient content in their soil. In fact, over-fertilizing can lead to root rot or attract pests. A balanced, low-nitrogen fertilizer can be added sparingly during the growing season, but only if the plant is not getting sufficient nutrients naturally.

4. Additives: Some gardeners add horticultural sand, perlite, or vermiculite to their cactus soil to improve drainage. These additives help to loosen the soil and increase porosity, allowing excess water to drain away quickly.

5. Soil Mix: A suitable soil mix for cacti can be made by mixing one part sand or perlite with two parts coarse-textured potting soil. This mixture should be well-draining and allow excess water to escape easily.

In summary, cacti require a well-draining, porous soil mix that is low in organic matter and adequately textured to promote drainage. A balanced, low-nitrogen fertilizer can be added sparingly during the growing season, but only if the plant is not getting sufficient nutrients naturally. By providing the right soil conditions, you can help ensure healthy growth and longevity for your cacti.

---

**PFP response:**
Cactus prefer well-draining soil that is sandy, porous, and low in organic matter. A cactus soil mix should be made up of a combination of coarse sand, perlite, peat moss, andor vermiculite. The ideal soil pH for cactus is between 6.0 and 7.0. Avoid using heavy clay soils or organic-rich soils as they can retain too much moisture and lead to root rot. It is also important to provide good drainage by using a container with drainage holes or by placing a layer of gravel or pebbles at the bottom of the container.

---

Figure 12: **Response in AlpacaEval2.0 .** For a qualitative comparison between Iterative DPO and PFP, we generate the responses from the same prompt.

Table 8: **Evaluation on Arena-Hard-Auto.** Evaluation results on Arena-Hard-Auto with different variants of Mistral-7B-v0.1. The best scores are highlighted in **bold**.

| Methods | Score | Avg. len |
|---|---|---|
| SFT | 2.2 | 507 |
| Iterative DPO | 4.4 | 671 |
| SELFEE | **9.0** | 839 |
| PFP (Ours) | 7.0 | **463** |

Only the batch size was set to 32, the same as our setup, and the remaining hyperparameters from SimPO's official github were used as is.[5] The results are presented in Table 9, and one can observe that PFP yields better alignment performance with less length bias.

---

[5] https://github.com/princeton-nlp/SimPO

---

**Prompt:**
Who is Larry Page?

---

**Iterative DPO response:**
Larry Page is an American entrepreneur, computer scientist, and co-founder (along with Sergey Brin) of Google, one of the largest technology companies in the world. He was born on March 26, 1973, in East Lansing, Michigan, and grew up in the area.

Page earned his Bachelor's degree in Computer Science and Engineering in 1995 from the University of Michigan, where he later received an honorary Doctor of Science degree in 2005. After completing his studies, Page moved to Stanford University, where he pursued his PhD in Computer Science.

Page and Brin founded Google in 1998, initially as a graduate school project, with the aim of building a better search engine than those available at the time. The company quickly gained popularity and became a major player in the tech industry. Page served as Google's Chief Executive Officer (CEO) from 1999 until 2001, when he handed the reins over to Eric Schmidt.

Page has continued to play an active role in Google's development, serving as the company's CEO again from 2011 until 2019, when he stepped down in favor of Sundar Pichai. He currently serves as the CEO of Alphabet Inc., the parent company of Google, and continues to be a key figure in the company's direction and innovation.

Page is known for his focus on long-term thinking, his commitment to sustainability and renewable energy, and his dedication to improving the quality of life for people around the world through technology. He has also been recognized for his philanthropic efforts, including his work to improve education and health outcomes in developing countries through the Larry Page Center for Entrepreneurial Studies and the Page Cancer Center at Stanford University.

---

**PFP response:**
Larry Page is a co-founder of Google, a multinational technology company that provides search engine services, online advertising, and other internet-related products and services. He was born on March 26, 1973, in East Lansing, Michigan, and graduated from Stanford University with a degree in computer science and artificial intelligence. Page is known for his innovative ideas, leadership, and vision in the technology industry. He serves as the CEO of Alphabet Inc., the parent company of Google, and is one of the richest people in the world. Page is also known for his philanthropic efforts and supports various causes, including healthcare research and education.

---

Figure 13: **Response in AlpacaEval2.0 .** For a qualitative comparison between Iterative DPO and PFP, we generate the responses from the same prompt.

Table 9: **Comparison with SimPO.** Evaluation results on AlpacaEval 2.0 with different variants of Mistral-7B-v0.1. The best scores are highlighted in **bold**.

| Methods | LC Win Rate | Win Rate | Avg. len |
|---|---|---|---|
| DPO | 9.93 | 8.02 | 1409 |
| SimPO | 9.29 | **10.94** | 2192 |
| PFP (Ours) | **15.24** | 10.18 | **1187** |

**Robustness with potential variance.** Since our evaluation using AlpacaEval 2.0 rely on GPT-4 as the evaluator, this can potentially raise a question about the variability in the results due to using GPT-4 for evaluation, particularly since its responses can introduce variance. To address this, we conducted two additional rounds of experiments. In these new experiments, we fixed the initial DPO model and repeated subsequent evaluations to assess consistency. We conducted an evaluation on AlpacaEval 2.0, and the results are presented in Table 10. While the differences in LC Win Rate may appear modest (*e.g.*, 1.46% improvement), the variance for these evaluations (*e.g.*, 0.29 for PFP) supports the statistical significance of these results. Moreover, we remark that the primary contribution of our method lies in feature debiasing. As shown in Figure 4, PFP demonstrates substantial improvements over other baselines in mitigating preference feature bias.

---

**Prompt:**
What causes the northern lights

---

**Iterative DPO response:**
The Northern Lights, also known as Aurora Borealis, are caused by the interaction of charged particles from the sun (solar wind) with the Earth's magnetic field and atmosphere.

When the solar wind collides with the Earth's magnetic field, it creates a region called the magnetosphere. This region traps some of the charged particles from the solar wind, including electrons and protons.

When these charged particles enter the Earth's atmosphere, they collide with the neutral atoms and molecules that make up the Earth's atmosphere, causing them to become electrically charged and emit light. The colors of the Northern Lights are determined by the type of atoms and molecules that are being ionized, as well as the altitude and energy of the collisions.

The most common colors of the Northern Lights are green, yellow, and red, but other colors such as blue, purple, and orange can also be seen under certain conditions. The brightness and movement of the Northern Lights depend on the intensity of the solar wind and the strength of the Earth's magnetic field at that particular location and time.

---

**PFP response:**
The northern lights, also known as aurora borealis, are caused by the interaction between charged particles from the sun (primarily electrons and protons) and the Earth's magnetic field and atmosphere. When these charged particles enter the Earth's magnetic field, they are deflected towards the polar regions. As they collide with the Earth's atmosphere, they excite and ionize the gas molecules, causing them to emit light in the form of colorful streaks and shapes. The colors of the northern lights depend on the type of gas molecule that is being excited and the energy of the particles colliding with them. The most common colors are green, yellow, red, blue, and purple. The intensity and frequency of northern lights depend on the solar wind, the strength and direction of the Earth's magnetic field, and other geomagnetic factors.

---

Figure 14: **Response in AlpacaEval2.0 .** For a qualitative comparison between Iterative DPO and PFP, we generate the responses from the same prompt.

Table 10: **Robustness on potential variance.** Evaluation results on AlpacaEval 2.0 with different random seeds.

| Methods | 1st Seed Data | 2nd Seed Data | 3rd Seed Data | Average | Variance |
|---|---|---|---|---|---|
| PFP: LC Win Rate (%) | 15.24 | 14.38 | 14.22 | 14.61 | 0.29 |
| PFP: Win Rate (%) | 10.18 | 10.16 | 9.97 | 10.10 | 0.01 |
| SELFEE: LC Win Rate (%) | 14.23 | 12.58 | 12.64 | 13.15 | 0.84 |
| SELFEE: Win Rate (%) | 17.49 | 15.59 | 17.06 | 16.71 | 0.99 |

Table 11: **Simple system prompt for length bias.** Evaluation results on AlpacaEval 2.0 with different variants of Mistral-7B-v0.1. The best scores are highlighted in **bold**.

| Methods | LC Win Rate | Win Rate | Avg. len |
|---|---|---|---|
| SELFEE | 14.23 | **17.49** | 2412 |
| SELFEE + Concise | 13.40 | 15.50 | 2218 |
| PFP (Ours) | **15.24** | 10.18 | **1187** |

**Simple system prompt to mitigate length bias.** We further conduct the new experiment by adding "being concise" in the system prompt, as another baseline (*SELFEE + Concise*). The results are presented in Table 11, and we found that it led to some reduction in response length, but it also resulted in decreased overall performance.

**Preference feature distribution.** Here, we present the preference feature distributions specifically. For each category of preference feature, we normalize the fre-

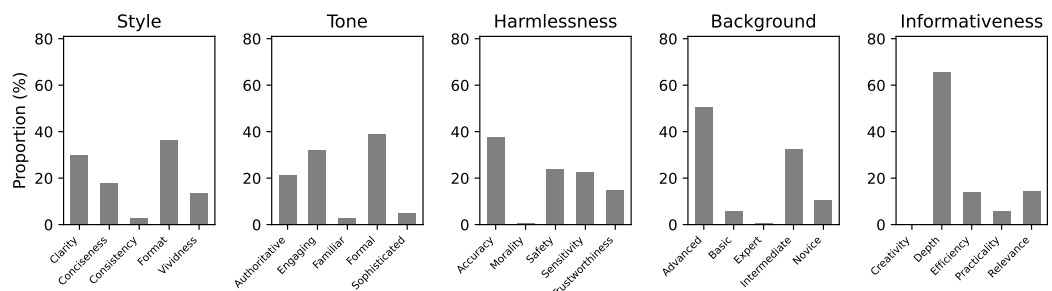

Figure 16: **Preference feature distribution captured in seed dataset**

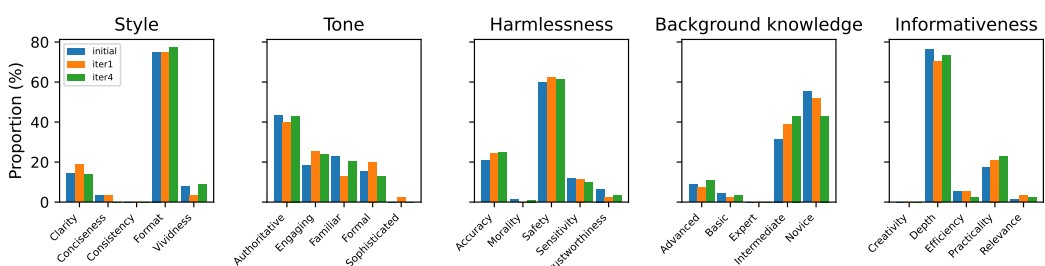

Figure 17: **Preference feature distribution captured in responses generated from PFP**

quency and present the proportion of each sub-feature. Figure 16 is the distribution of seed preference dataset, which is extracted with feature extractor (see Section 4.1). Remarkably, one can observe the imbalanced distribution for each category, which potentially affect to the classifier's performance. Next, in Figures 17, 18, 19, we present the preference feature distribution under different online preference learning methods. Unlike Figure 16, this feature is measured by a single response generated from the AlpacaEval 2.0 prompt. Among all preference features, we select the feature with the largest change under each method and present them in Figure 15. Here, it is clearly observed that PFP yields much smaller change in preference feature, compared to SELFEE and Iterative DPO. We note that the overall tendency of change can be also verified in Figure 4.

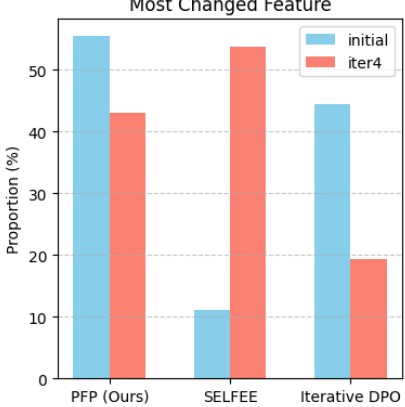

Figure 15: **Distribution of most changed feature.**

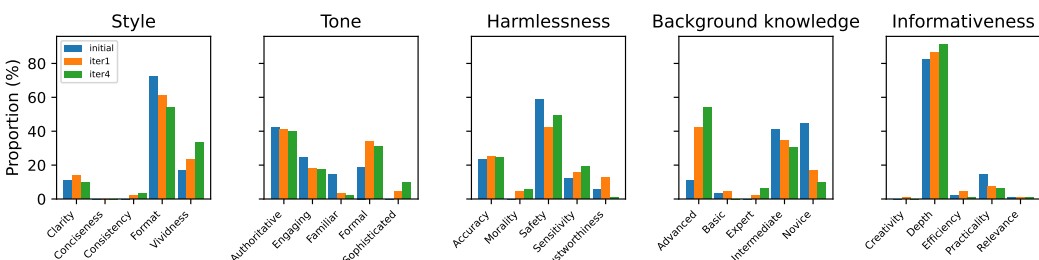

Figure 18: **Preference feature distribution captured in responses generated from SELFEE**

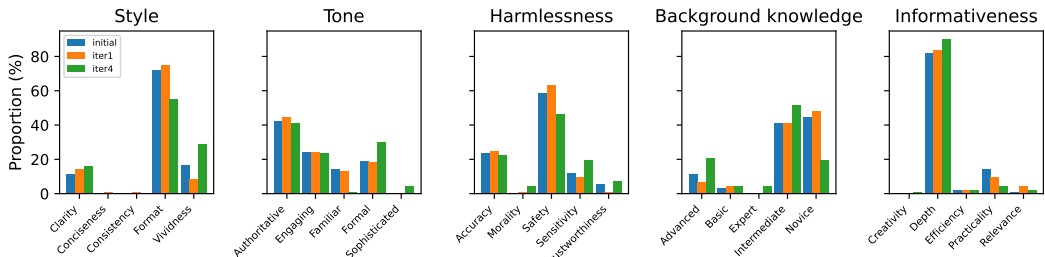

Figure 19: **Preference feature distribution captured in responses generated from Iterative DPO**

