# OpenReview forum: "Debiasing Online Preference Learning via Preference Feature Preservation"
_ICLR.cc/2025/Conference — Submitted to ICLR 2025_

### Official Review · Reviewer_VhEv · 2024-11-01

**Soundness:** 2
**Presentation:** 1
**Contribution:** 2
**Rating:** 5
**Confidence:** 3

**Summary:**

This paper introduces a method to mitigate biases that arise during LM alignment -- focusing specifically on online DPO. Authors propose a new framework (PFP) to maintain the ‘distribution of human preferences’ throughout the training process, as they argue that it usually shifts during online DPO optimization.

**Strengths:**

- Addressing biases underlying human preference data is an important problem.
- Experimental results are strong, and the approach is interpretable.

**Weaknesses:**

1. The motivation for the proposed method is unclear, and its description often confusing. Examples:
  - On what basis can we assume to distill the full complexity of human preferences down to discrete unsupervised feature dimensions?
  - What is the motivation behind the process of distribution preservation (paragraph starting L254)?
  - Paragraph starting L173: i am very familiar with this literature but find this explanation confusing. Missing related works here: mention SLiC-HF (Zhao et al., 2023) and online DPO (Calandriello et al., 2023).
  - L307: how can you change the system prompt to be only one of $(s_1, s_2)$? Surely the response which does not correspond to the chosen system prompt is going to be worse?
 - Importantly, the method proposed involves many modeling choices that are not properly ablated, which makes it tricky to know whether all are needed. Some of the following modeling steps are ablated but most are not thoroughly evaluated:
   - Quality of the feature classifier
   - Performance of the distribution matching step
   - Synthesizing system prompt from preference features (see positioning wrt related works below)
   - Double system prompt sampling (see Q above)
   - Curriculum learning via temperature scheduling
  - examples of grammatical issues/ typos:
    - L147: beginning of the sentence
    - L246
    - L248: 'auxiliarly'

2. The experimental setup is weak.
- Weak positioning wrt. rest of the literature. For example, why not compare to prompt optimization strategies such as OPRO (Yang et al., 2023) and RLCD (Yang et al 2024)?
- Since the motivation is to avoid biases, this method requires rigorous evaluation on a task with specific biases... Here we only measure for length bias. Could authors not consider an experimental setting with a clearer bias? Does your approach then accentuate biases present in the preference data?

**Questions:**

See above.

---

> ### Author Response · Authors · 2024-11-22
> **Response to Reviewer VhEv (1/2)**
>
> Dear Reviewer VhEv,
>
> We sincerely appreciate your thoughtful comments. We have carefully considered each of your questions and provide detailed responses below.
>
> ---
>
> **[W1-1] On what basis can we assume to distill the full complexity of human preferences down to discrete unsupervised feature dimensions?**
>
> Thank you for raising this important point about the complexity of human preferences. We agree that human preferences are multifaceted and not easily reduced to discrete feature dimensions. However, our approach does not aim to capture the entirety of human preference complexity. Instead, we focus on mitigating the unintended biases that arise during online preference learning by preserving the distribution of the key preference features that can be extracted in existing datasets.
>
> The feasibility of extracting and utilizing preference features has been supported by prior research. For example, the approach of deriving preference features from two labeled responses has been validated experimentally [1]. Additionally, studies like Janus [2] have shown that even predefined discrete features can effectively capture and represent a wide range of human preferences when used for supervised fine-tuning. Building on this foundation, we designed our framework to use a more extensive set of predefined features, enhancing its capacity to reflect diverse human preferences. These prior successes underscore the practicality and validity of our method in leveraging discrete feature dimensions to mitigate biases in online preference learning.
>
> ---
>
> **[W1-2] What is the motivation behind the process of distribution preservation (paragraph starting L254)?**
>
> The distribution preservation process is introduced to ensure that the distribution of preference features assigned to a new input prompt set aligns with the human preference feature distribution observed in the initial seed dataset. This is necessary because directly predicting preference features using the trained feature classifier (lines 247-252) can be insufficient due to the classifier’s limitations (see Table 7 of the revised draft), such as being trained on a small dataset and the long-tailed nature of preference features (see Figure 15 of the revised draft).
>
> We also note that the effectiveness of this process is demonstrated in Figure 5(b), where the KL divergence between the human preference feature distribution and the assigned distribution is significantly reduced when using distribution preservation compared to naive predictions. This alignment leads to improved performance, including better alignment scores and reduced length bias, as shown in Table 2.
>
> ---
>
> **[W1-3] Paragraph starting L173: i am very familiar with this literature but find this explanation confusing. Missing related works here: mention SLiC-HF (Zhao et al., 2023) and online DPO (Calandriello et al., 2023)**
>
> We appreciate the constructive comments to improve the clarity of the draft and the suggestion to include additional related work. First, we have improved the paragraph to resolve the confusion in the revised draft (lines 192-196 of the revised draft). Next, we have added the references to SLiC-HF and online DPO in Section 2 of the revised draft.
>
> ---
>
> **[W1-4] L307: how can you change the system prompt to be only one of (s1,s2)? Surely the response which does not correspond to the chosen system prompt is going to be worse?**
>
> We randomly sample the one system prompt between $(s_1,s_2)$. After that, we consider the response from the sampled system prompt as a preferred response $y_w$ under this system prompt, and the other response which does not correspond to this system prompt as a dispreferred response $y_l$.

---

> ### Author Response · Authors · 2024-11-22
> **Response to Reviewer VhEv (2/2)**
>
> **[W1-5] Importantly, the method proposed involves many modeling choices that are not properly ablated, which makes it tricky to know whether all are needed. Some of the following modeling steps are ablated but most are not thoroughly evaluated: (1) Quality of the feature classifier, (2) Performance of the distribution matching step, (3) Synthesizing system prompt from preference features (see positioning wrt related works below), (4) Double system prompt sampling (see Q above), (5) Curriculum learning via temperature scheduling**
>
> We clarify that most of the modeling steps mentioned, except for (1) the quality of the feature classifier, have been evaluated in our experiments. Specifically:
> (2) Performance of the distribution matching step and (3) synthesizing system prompts from preference features are analyzed in Table 2 and Figure 5(b).
> (4) Double system prompt sampling and (5) curriculum learning via temperature scheduling are evaluated in Table 3 and Figure 5(c).
>
> For (1), we have conducted additional experiments to measure the accuracy of the trained feature classifiers on a separate test dataset. The results show moderate accuracy, which is limited by the small amount of training data and the long-tailed nature of preference features (see Figure 15 in the revised draft). These findings highlight the importance of the proposed distribution preservation step to address the feature classifier’s limitations. We have included these results and their discussion in Appendix F of the revised draft.
>
> \begin{array}{c|ccccc}
> \hline
> \text{Metric} & \text{background} & \text{harmlessness} & \text{informativeness} & \text{style} & \text{tone} \newline \hline
> \text{accuracy} & 0.535 & 0.512 & 0.688 & 0.496 & 0.507 \newline
> \text{f1} & 0.532 & 0.513 & 0.663 & 0.489 & 0.426 \newline \hline
> \end{array}
>
> ---
>
> **[W2-1] Weak positioning wrt. rest of the literature. For example, why not compare to prompt optimization strategies such as OPRO and RLCD?**
>
> We clarify that our work is distinct from prompt optimization research like OPRO and RLCD. Prompt optimization aims to maximize output quality by finding the most effective prompts through systematic tuning. In contrast, our approach focuses on ensuring that system prompts reflect specific preference features during response generation or preference labeling, for preserving nuanced human preferences rather than simply maximizing output performance.
>
> ---
>
> **[W2-2] Since the motivation is to avoid biases, this method requires rigorous evaluation on a task with specific biases... Here we only measure for length bias. Could authors not consider an experimental setting with a clearer bias? Does your approach then accentuate biases present in the preference data?**
>
> Thank you for the opportunity to clarify. Beyond addressing length bias, we conducted experiments to examine biases in broader preference features, as shown in Figures 4 and 5. Specifically, we measured the KL divergence between the preference feature distributions assigned to new input prompts and the human preference feature distribution from the seed dataset. An increasing KL divergence across iterations indicates growing biases as the assigned features deviate from the original distribution. The results in Figures 4 and 5 show that the proposed PFP effectively mitigates these biases (i.e., maintaining low KL divergence), while baselines such as iterative DPO and SELFEE fail to do so. This demonstrates PFP’s efficacy in addressing biases across multiple preference feature aspects during online preference learning.
>
> ---
>
> **[W3] Typo**
>
> Thanks for your careful reading. We have corrected these typos in the revised draft.
>
> ---
>
> [1] Pitis et al., Improving context-aware preference modeling for language models., arXiv:2407
> [2] Lee et al., Aligning to thousands of preferences via system message generalization., NeurIPS 2024
>
> ---
>
> If you have any further questions/concerns, please do not hesitate to let us know.
>
> Thank you very much,
> Authors

---

> > ### Comment · Reviewer_VhEv · 2024-11-25
> >
> > Thank you for your rebuttal. While I appreciate the clarifications, my concerns about the weakness of the experimental setup remain, with modeling choices and bias mitigation still lacking a rigorous ablation. In fact, since you mention additional papers doing bias mitigation for RLHF, I would like to point out that these have a much more extensive and convincing experimental setup, quantifying these biases beyond length measurements or KL divergences against unsupervised features, as well as across a wider range of models and datasets.

---

> > > ### Author Response · Authors · 2024-11-27
> > > **Response to Reviewer VhEv**
> > >
> > > Dear Reviewer VhEv,
> > >
> > > We appreciate your response to our rebuttal, but we would like to clarify that our works aims to mitigate biases related to preference features. As other reviewers have pointed out (PYLd, sywP), our approach proposes a new problem and a novel way to address it compared to existing methodologies, and there are currently NO directly comparable experimental setups in the literature. Specifically, our evaluation was designed to accurately assess our objectives in bias mitigation using KL Divergence, and additionally, we conducted a detailed examination of length bias. Given the unique nature of our debiasing approach, we consider our experimental setup sufficient for demonstrating the efficacy of our proposed methods.
> > >
> > > If you have any further questions/concerns, please do not hesitate to let us know.
> > >
> > > Thank you very much,
> > > Authors

---

### Official Review · Reviewer_sywP · 2024-11-03

**Soundness:** 3
**Presentation:** 3
**Contribution:** 3
**Rating:** 6
**Confidence:** 3

**Summary:**

The paper presents a novel framework, PFP (Preference Feature Preservation), which addresses the issue of bias in large language models (LLMs) during online preference learning. The core content revolves around the innovative approach of maintaining the distribution of human preference features throughout the online learning process. This is achieved by training a feature classifier on existing offline pairwise human preference data, mapping appropriate preference features for each input instruction during online learning, and incorporating these features into system prompts for LLM training. The experiments indicate that PFP successfully mitigates bias in preference features and outperforms previous methods on general benchmarks like AlpacaEval 2.0 and MT-Bench, also nearly resolving the length bias issue.

**Strengths:**

1. Originality: The introduction of the PFP framework is a novel approach to addressing bias in online preference learning for large language models (LLMs). This approach to nearly resolving the length bias issue, which has been a long-standing problem in online preference learning.
2. Rigorous Experimental Design: The paper presents a well-structured set of experiments that validate the effectiveness of the PFP framework. The use of established benchmarks like AlpacaEval 2.0 and MT-Bench adds to the credibility of the results.
3. Significance: The paper's contribution to reducing bias in AI systems is significant. By addressing bias in LLMs, the research has implications for the ethical deployment of AI, which is a critical concern in the field.

**Weaknesses:**

1. Diversity of tasks: The paper primarily uses AlpacaEval 2.0 and MT-Bench for evaluation. While these are established benchmarks, the use of additional or more diverse datasets could strengthen the claims of the framework's effectiveness. For example, the preference features of math or coding tasks may be different, the author should give more insights on various tasks.
2. Comparative Analysis with State-of-the-Art Methods: The paper compares PFP with SFT, DPO and Iterative DPO but does not include a comparison with the latest state-of-the-art methods in bias mitigation for LLMs. Including comparisons with cutting-edge preference learning methods would provide a clearer picture of PFP's performance relative to the most advanced techniques.

**Questions:**

The paper assumes a predefined set of preference features based on certain definitions and classifications. These assumptions may not cover the full spectrum of human preferences, and the preference features of math or coding tasks may be different. How to choose the most proper preference features for different tasks?

---

> ### Author Response · Authors · 2024-11-22
> **Response to Reviewer sywP (1/2)**
>
> Dear Reviewer sywp,
>
> We sincerely appreciate your thoughtful comments. We have carefully considered each of your questions and provide detailed responses below.
>
> ---
>
> **[W1] Diversity of tasks: Use of additional or more diverse datasets could strengthen the claims of the framework's effectiveness.**
>
> We clarify that our framework primarily targets modeling common human preferences in general language tasks. While math and coding tasks are important domains, they require specialized preference features that differ fundamentally from common human preferences. These specialized domains often use distinct data collection methods and training approaches, as demonstrated in recent works [1,2]. Given our framework’s specific focus on general human preference alignment, including these specialized domains would fall outside our intended scope.
>
> Nevertheless, we agree that evaluating our framework on more diverse datasets can provide a stronger assessment of our framework. Following your suggestion, we have additionally evaluated our framework and key baseline methods on the "Arena-Hard-Auto" benchmark [3], which is another well-established benchmark to evaluate LLM alignment [4]. We present the results on the below table. Here, one can observe that while PFP performed slightly below SELFEE, it still outperforms the iterative DPO model by a significant margin. Notably, PFP consistently produced shorter responses compared to both SELFEE and iterative DPO, indicating effective bias mitigation in response length. We have added these new results and the discussion in Appendix F of the revised draft.
>
> \begin{array}{l|cc}
> \hline
> \text{Methods} & \text{Score} & \text{Avg. len} \newline \hline
> \text{SFT} & 2.2 & 507 \newline
> \text{Iterative DPO} & 4.4 & 671 \newline
> \text{SELFEE} & \textbf{9.0} & 839 \newline \hline
> \text{PFP (Ours)} & 7.0 & \textbf{463} \newline \hline
> \end{array}
>
> ---
>
> **[W2] Comparative Analysis with State-of-the-Art Methods: The paper compares PFP with SFT, DPO and Iterative DPO but does not include a comparison with the latest state-of-the-art methods in bias mitigation for LLMs. Including comparisons with cutting-edge preference learning methods would provide a clearer picture of PFP's performance relative to the most advanced techniques.**
>
> We first clarify that our work is the first work that identifies the preference feature bias during the iterative preference learning; therefore, there are no comparable baselines that can be directly applied to mitigate this problem.
>
> Next, in the perspective of length bias, we remark that the comparison with the latest state-of-the-art methods is already included in the original draft; in Tables 4 and 5, we observed that the proposed PFP is significantly better to mitigate the length bias compared to two different state-of-the-art methods based on the length penalty in reward model [5] and modified DPO objective [6].
>
> Lastly, to respond to your suggestion to include cutting-edge preference learning methods, we have additionally conducted the experiments with SimPO [4], the latest offline preference alignment algorithm, as our new baseline. This baseline is trained using the same seed data size as PFP, which is also the same amount of data used by DPO. Only the batch size was set to 32, the same as our setup, and the remaining hyperparameters from Simpo's official github were used as is. The results are presented below, and one can observe that PFP yields better alignment performance with less length bias.
>
> \begin{array}{l|ccc}
> \hline
> \text{Methods} & \text{LC Win Rate} & \text{Win Rate} & \text{Avg. len} \newline \hline
> \text{DPO} & 9.93 & 8.02 & 1409 \newline
> \text{SimPO} & 9.29 & \textbf{10.94} & 2192 \newline \hline
> \text{PFP (Ours)} & \textbf{15.24} & 10.18 & \textbf{1187} \newline  \hline
> \end{array}

---

> ### Author Response · Authors · 2024-11-22
> **Response to Reviewer sywP (2/2)**
>
> **[Q1] The paper assumes a predefined set of preference features based on certain definitions and classifications. These assumptions may not cover the full spectrum of human preferences, and the preference features of math or coding tasks may be different. How to choose the most proper preference features for different tasks?**
>
> Our approach does not aim to choose the proper preference features for different tasks. Instead, we focus on mitigating the unintended biases that arise during online preference learning by preserving the distribution of the key preference features that can be extracted in existing datasets. Therefore, we used the fixed set of preference features through all the experiments, and the other categories of preference features can be also considerable.
>
> The feasibility of the current way to extract and utilize preference features has been supported by prior research. For example, the approach of deriving preference features from two labeled responses has been validated experimentally [7]. Additionally, studies like Janus [8] have shown that even predefined discrete features can effectively capture and represent a wide range of human preferences when used for supervised fine-tuning. Building on this foundation, we designed our framework to use a more extensive set of predefined features, enhancing its capacity to reflect diverse human preferences. These prior successes underscore the practicality and validity of our method in leveraging discrete feature dimensions to mitigate biases in online preference learning.
>
> ---
>
> [1] Yuan et al., Advancing llm reasoning generalists with preference trees., arXiv:2404
> [2] Mitra et al., Orca-math: Unlocking the potential of slms in grade school math., arXiv:2402
> [3] Li, et al., From Crowdsourced Data to High-Quality Benchmarks: Arena-Hard and BenchBuilder Pipeline., arXiv:2406
> [4] Meng et al., SimPO: Simple Preference Optimization with a Reference-Free Reward., NeurIPS 2024
> [5] Dong et al., RLHF Workflow: From Reward Modeling to Online RLHF., arXiv:24.05
> [6] Park et al., Disentangling Length from Quality in Direct Preference Optimization., Findings of ACL 2024
> [7] Pitis et al., Improving context-aware preference modeling for language models., arXiv:2407
> [8] Lee et al., Aligning to thousands of preferences via system message generalization., NeurIPS 2024
>
> ---
>
> If you have any further questions/concerns, please do not hesitate to let us know.
>
> Thank you very much,
> Authors

---

### Official Review · Reviewer_PYLd · 2024-11-03

**Soundness:** 2
**Presentation:** 2
**Contribution:** 2
**Rating:** 5
**Confidence:** 3

**Summary:**

The paper proposes a novel framework, PFP (Preference Feature Preservation), aimed at mitigating biases that arise in the preference learning process of LLMs.

**Strengths:**

The paper introduces a unique approach, Preference Feature Preservation (PFP), for managing bias in preference learning. By explicitly incorporating preference features in the system prompts and maintaining feature distribution, it provides a fresh angle on bias mitigation that has not been explored in existing work.

**Weaknesses:**

The paper introduces a set of predefined preference features, categorizing them into five distinct classes, which provides a structured framework for evaluating human preferences in various dimensions. However, in the main results, the experiments appear to primarily focus on addressing the length bias issue, leaving it unclear whether similar attention was given to the other identified preference classes. Were any experiments conducted to examine these additional preference aspects?

Furthermore, the evaluation results, including those from AlpacaEval and MT-Bench, rely on GPT-4 as the evaluator. This raises a question about the potential variability in the results due to using GPT-4 for evaluation, particularly since its responses can introduce variance. For the results shown in the paper, i didn't see the significant improvement compared with other baselines, only 1%~2% difference for AlpacaEval, pretty close for MT-Bench.

**Questions:**

Please see the weakness part.

---

> ### Author Response · Authors · 2024-11-22
> **Response to Reviewer PYLd**
>
> Dear Reviewer PYLd,
>
> We sincerely appreciate your thoughtful comments. We have carefully considered each of your questions and provide detailed responses below.
>
> ---
>
>  **[W1] Were any experiments conducted to examine these additional preference aspects?**
>
> Thank you for the opportunity to clarify. Beyond addressing length bias, we conducted experiments to examine biases in broader preference features, as shown in Figures 4 and 5. Specifically, we measured the KL divergence between the preference feature distributions assigned to new input prompts and the human preference feature distribution from the seed dataset. An increasing KL divergence across iterations indicates growing biases as the assigned features deviate from the original distribution. The results in Figures 4 and 5 show that the proposed PFP effectively mitigates these biases (i.e., maintaining low KL divergence), while baselines such as iterative DPO and SELFEE fail to do so. This demonstrates PFP’s efficacy in addressing biases across multiple preference feature aspects during online preference learning.
>
> ---
>
> **[W2] Potential variability in the results due to using GPT-4 for evaluation.**
>
> We acknowledge the reviewer’s concerns regarding the use of GPT-4 for evaluation and the variance it may introduce. To address this, we conducted two additional rounds of experiments. In these new experiments, we fixed the initial DPO model and repeated subsequent evaluations to assess consistency. We conducted an evaluation on AlpacaEval 2.0, and obtained the results below. While the differences in LC Win Rate may appear modest (e.g., 1.46% improvement), the variance for these evaluations (e.g., 0.29 for PFP) supports the statistical significance of these results. Moreover, we remark that the primary contribution of our method lies in feature debiasing. As shown in Figure 4, PFP demonstrates substantial improvements over other baselines in mitigating preference feature bias. We have added these new results in Appendix F of the revised draft.
>
> \begin{array}{l|ccc|cc}
> \hline
> \text{Methods} & \text{1st Seed Data} & \text{2nd Seed Data} & \text{3rd Seed Data} & \text{Average} & \text{Variance} \newline \hline
> \text{PFP (Ours): LC Win Rate} & 15.24 & 14.38 & 14.22 & 14.61 & 0.29 \newline
> \text{PFP (Ours): Win Rate} & 10.18 & 10.16 & 9.97 & 10.10 & 0.01 \newline \hline
> \text{SELFEE: LC Win Rate} & 14.23 & 12.58 & 12.64 & 13.15 & 0.84 \newline
> \text{SELFEE: Win Rate} & 17.49 & 15.59 & 17.06 & 16.71 & 0.99 \newline \hline
> \end{array}
>
> ---
>
> If you have any further questions/concerns, please do not hesitate to let us know.
>
> Thank you very much,
> Authors

---

> ### Author Response · Authors · 2024-12-01
> **Gentle reminder to Reviewer PYLd: The interactive discussion phase will end in less than two days**
>
> Dear Reviewer PYLd,
>
> We once again thank you for your review of our manuscript. We sincerely appreciate your valuable insights and feedback.
>
> As the discussion period is drawing to a close, we kindly remind you that there are two days remaining for further comments or questions. We would be grateful for the opportunity to address any additional concerns you may have before the discussion phase ends.
>
> Thank you for your time and consideration.
>
> Warm regards,
> Authors

---

### Official Review · Reviewer_639N · 2024-11-04

**Soundness:** 3
**Presentation:** 3
**Contribution:** 2
**Rating:** 6
**Confidence:** 3

**Summary:**

This paper tackles the bias problem as the online preference alignment takes place. The paper is motivated by the bias towards lengthy responses and proposes a system-prompt engineering-based solution. Since system prompts can be combinatorial explosive space, they only try to learn certain attributes (tone, informativeness, etc) and subattributes within them. This makes every instruction and response pair a function of 5 sub-attributes in total. Now, given the input and attribute data, they learn a mapping, which predicts the attributes that can lead to favoring certain responses.

Having learned this distribution, to create a new dataset for alignment, they first calibrate it with the prior distribution (average distribution over these attributes) for the given training data. Followed by calibration, they generate the new system prompts and then align the model.

Several ablation studies are conducted to show the efficacy of the proposed method.

**Strengths:**

I like the number of ablations that are being performed, and the comparison with other length-controlled generation-based baselines. The experimental setup seems to be complete.

**Weaknesses:**

Following are my questions that could improve this paper.

- Notations in this paper can be heavily improved, in particular for the FE part. I think one should use vector notation for the label space, and simplex to denote output of the FE network.
- I am still not fully sure about the motivation of this work. Can authors highlight cases where certain reward models prefer lengthy responses despite having incorrect answers? In my belief, as long as the answer is correct, and if the system prompt doesn't have instruction to be precise (succinct), then there is nothing wrong in long generations (other than the computation aspect)
- How is length-controlled generation done?
- Why should SELFEE work? To my understanding, it is some sort of pseudo-labeling based on its current state. If the model is not good enough to begin with, then wouldn't it exacerbate the biases (or if it is incorrect for certain prompts)?
- I am not sure why the output adjustment is needed. Can authors perform a study showing its use?
- In Fig 4, why is it bad that the distribution for the preference feature is changing (that is KL divergence increasing). I am not fully sure why should that be linked with longer-generation
- Can authors run an experiment that just adds "being concise" in the system prompt, as another baseline?

**Questions:**

Refer to the weakness.

Post rebuttal: I’m increasing my scores.

---

> ### Author Response · Authors · 2024-11-22
> **Response to Reviewer 639N (1/2)**
>
> Dear Reviewer 639N,
>
> We sincerely appreciate your thoughtful comments. We have carefully considered each of your questions and provide detailed responses below.
>
> ---
>
> **[W1] Notations in this paper can be heavily improved, in particular for the FE part**
>
> Thanks for your constructive comments! Following your suggestion, we have modified the corresponding parts in the revised draft (lines 233-235), as follows:
> - (**Original**) Specifically, for the input instruction $x$ and the two responses $y_{w}$ and $y_{l}$, the feature extractor is defined as $\mathbf{p} = LLM_{FE}(x,y_{l},y_{w})$ where $\mathbf{p}=[p_{1},...,p_{5}], p_i \in C_i$.
> - (**Revised**) Specifically, for the input instruction $x$ and the two responses $y_{w}$ and $y_{l}$, the feature extractor is defined as $\mathbf{p} = LLM_{\text{FE}}(x,y_l,y_w)$ where $\mathbf{p}=[p_{1},...,p_{5}]$, where each $p_i$ represents a probability distribution over the 5 sub-features of class $C_i$ (i.e., $p_i \in [0,1]^5$ and $\sum_{j=1}^5 p_i^j = 1$).
>
> ---
>
> **[W2] I am still not fully sure about the motivation of this work… In my belief, as long as the answer is correct, and if the system prompt doesn't have instruction to be precise (succinct), then there is nothing wrong in long generations (other than the computation aspect)**
>
> First, we would like to clarify that our main motivation is to eliminate biases at the preference feature level, rather than simply addressing length bias. Indeed, the reducing length bias is a side effect from debiasing at the preference feature level.
>
> Next, regarding the problem with lengthy responses, length bias is a form of reward over-optimization that has been persistently noted as an issue [1,2,3,4]. While response length does show some correlation with human preference, current reward modeling tends to favor overly lengthy responses, leading to a divergence from actual human preferences. As a result, these models often generate responses that stray away from true human preferences, and therefore it is undesirable. But, it’s true that the generation of lengthy responses itself can be not problematic if the superiority between responses could be objectively defined like coding or math tasks using the correctness. However, we note that the superiority of responses cannot be explicitly measured depending on the situation, such as in open-ended tasks or subjective domains.
>
> ---
>
> **[W3] How is length-controlled generation done?**
>
> We did not separately generate the length-controlled responses to measure length-controlled (LC) win rate. Indeed, LC win-rate and normal win-rate are simultaneously evaluated for the same responses using GPT-4 as evaluator, while LC win-rate is measured by neutralizing the effect of response length to focus on quality, using a separately trained regression model [1].
>
> ---
>
> **[W4] Why should SELFEE work? … If the model is not good enough to begin with, then wouldn't it exacerbate the biases (or if it is incorrect for certain prompts)?**
>
> We first remark that SELFEE begins with an LLM fine-tuned using DPO on the initial seed preference dataset; therefore, depending on the size of the seed dataset and the degree of distribution shift in new prompts for each iteration, the effectiveness of SELFEE can vary. However, our experiments (Table 1) show that it yields significant improvements in alignment performance, demonstrating its suitability for our setup. Additionally, as observed in Table 1 and Figure 4, even online preference learning with an external reward model (Iterative DPO) experiences biased preference features and increased response lengths. This indicates that the increased response length in SELFEE is not merely a result of exacerbating biases from an insufficiently strong starting model but reflects broader challenges inherent in current online preference learning methods. Lastly, we note that the proposed PFP is also based on SELFEE framework which is more vulnerable for the bias, but it successfully reduce the bias in preference features and response length.

---

> ### Author Response · Authors · 2024-11-22
> **Response to Reviewer 639N (2/2)**
>
> **[W5] I am not sure why the output adjustment is needed. Can authors perform a study showing its use?**
>
> The output adjustment process (lines 252-269 of the revised draft) is introduced to ensure that the distribution of preference features assigned to a new input prompt set aligns with the human preference feature distribution observed in the initial seed dataset. This is necessary because directly predicting preference features using the trained feature classifier (lines 247-252 of the revised draft) can be insufficient due to the classifier’s limitations (see Table 7 of the revised draft), such as being trained on a small dataset and the long-tailed nature of preference features.
> We also note that the effectiveness of this process is demonstrated in Figure 5(b), where the KL divergence between the human preference feature distribution and the assigned distribution is significantly reduced when using output adjustment compared to naive predictions. This alignment leads to improved performance, including better alignment scores and reduced length bias, as shown in Table 2.
>
> ---
>
> **[W6] In Fig 4, why is it bad that the distribution for the preference feature is changing (that is KL divergence increasing). I am not fully sure why should that be linked with longer-generation**
>
> An increasing KL divergence across iterations indicates growing biases in the preference feature distribution, as the assigned features deviate further from the original human preference distribution in the seed dataset. Since our goal is to minimize such biases, a higher KL divergence is undesirable. On the other hand, it’s true that this increased KL is NOT directly linked with longer-generation; however, we hypothesize length bias may arise because certain features correlated with response length, such as the depth of informativeness or user’s background knowledge. Therefore, the biases on these features across the iterations (see Figures 16-18 in Appendix F of the revised draft), potentially contributes to longer responses.
>
> ---
>
> **[W7] Can authors run an experiment that just adds "being concise" in the system prompt, as another baseline?**
>
> Thank you for the suggestion. Following your suggestion, we have conducted this experiment; we found that it (SELFEE + Concise) led to some reduction in response length, but it also resulted in decreased overall performance. We have added these results in Table 11 in Appendix F of the revised draft.
>
> \begin{array}{c|ccc}
> \hline
> \text{Methods} & \text{LC Win Rate} &\text{Win Rate} &\text{Avg. len}
> \newline \hline
> \text{SELFEE} & 14.23 & \textbf{17.49} & 2412 \newline
> \text{SELFEE + Concise} & 13.40 & 15.50 & 2218 \newline
> \text{PFP (Ours)} & \textbf{15.24} & 10.18 & \textbf{1187} \newline \hline
> \end{array}
>
> ---
>
> [1] Dubois et al., Length-controlled alpacaeval: A simple way to debias automatic evaluators., arXiv:2404
> [2] Singhal et al., A long way to go: Investigating length correlations in rlhf., arXiv:2310
> [3] Park et al., Disentangling length from quality in direct preference optimization., arXiv:2403
> [4] Yuan et al., Following length constraints in instructions., arXiv:2406
>
> ---
>
> If you have any further questions/concerns, please do not hesitate to let us know.
>
> Thank you very much,
> Authors

---

> > ### Comment · Reviewer_639N · 2024-11-25
> > **Quick question**
> >
> > - I thank the authors for providing new numbers, and intuition for the KL Divergence. I like the newly added figs 16-18. Perhaps they can add a figure showing which feature has the highest change from the initial after some rounds for all the methods, and by how much. This would help to understand the provided intuition; currently one has to look at all the bars and manually think of the difference.
> >
> > I have a quick question
> >
> > - On W5 "such as being trained on a small dataset and the long-tailed nature of preference features". If the dataset was hypothetically balanced, or not skewed towards certain classes, would that alleviate the need for further adjustment?

---

> > > ### Author Response · Authors · 2024-11-27
> > > **Response to Reviewer 639N**
> > >
> > > Dear Reviewer 639N,
> > >
> > > We appreciate your response to our rebuttal.
> > > - **Figure to summarize**: Following your suggestion, we have added Figure 15 to our new revision. Specifically, we select the feature with the largest change under each method and present them together. Here, it is clearly observed that PFP yields much smaller change in preference feature, compared to SELFEE and Iterative DPO. At the same time, we remark that the overall tendency of change can be also verified in Figure 4.
> > > - **Balanced distribution**: Even with hypothetically balanced classes, the complexity of feature classification and limited dataset size would likely still impose accuracy constraints. Therefore, although balanced data could potentially improve performance, we believe that the adjustment mechanisms would remain valuable tools for addressing residual limitations and optimizing model performance.
> > >
> > > If you have any further questions/concerns, please do not hesitate to let us know.
> > >
> > > Thank you very much,
> > > Authors

---

> > > > ### Author Response · Authors · 2024-12-01
> > > > **Gentle reminder to Reviewer 639N: The interactive discussion phase will end in less than two days**
> > > >
> > > > Dear Reviewer 639N,
> > > >
> > > > We once again thank you for your review of our manuscript. We sincerely appreciate your valuable insights and feedback.
> > > >
> > > > As the discussion period is drawing to a close, we kindly remind you that there are two days remaining for further comments or questions. We would be grateful for the opportunity to address any additional concerns you may have before the discussion phase ends.
> > > >
> > > > Thank you for your time and consideration.
> > > >
> > > > Warm regards,
> > > > Authors

---

### Comment · Area_Chair_T4N1 · 2024-11-25

Dear Reviewers,

This is a kind reminder that the dicussion phase will be ending soon on November 26th. Please read the author responses and engage in a constructive discussion with the authors.

Thank you for your time and cooperation.

Best,

Area Chair

---

### Meta-Review · Area_Chair_T4N1 · 2024-12-19

**Metareview:**

This paper proposes an online preference learning framework named Preference Feature Preservation (PFP). The PFP framework introduces fine-grained preference features and keeps the distribution of preference features consistent across multiple iterations to avoid the potential bias in the model. Experiments show that PFP can mitigate the bias in online preference learning to some extent.



However, the reviewers have pointed out some important weaknesses.

- First, the experiment settings (including tasks, datasets, benchmarks, and evaluation metrics) need significant improvements to show the effectiveness of PFP in reducing various biases. Except for the length bias evaluated using LC win rate and the length measurement, other types of bias need more rigorous and convincing evaluation metrics, not just the KL divergence between feature distributions.



- Second, the authors may want to conduct more comprehensive, rigorous, and adequate ablation studies than those in the current submission to provide insights into the various components of PFP, such as the feature classifier, the distribution matching step, and the sampling of double system prompts.



- Third, the comparisons between PFP and some related approaches (e.g., OPRO [1] and RLCD [2]) are missing. Although the authors claim that these methods are irrelevant to this paper, they do share some similarities with PFP, as they also discuss how system prompts affect the model's responses.





Therefore, I recommend the next version of this paper to incorporate more experiments and insights.



[1] Large Language Models as Optimizers.

[2] RLCD: Reinforcement Learning from Contrastive Distillation for LM Alignment.

**Additional Comments On Reviewer Discussion:**

This paper's initial average score was 5.25, and the initial scores given by the four reviewers were 6 (sywP), 5 (639N), 5 (VhEv), and 5 (PYLd). During the discussion phase, Reviewer 639N raised the score from 5 to 6. Finally, the scores of this paper are 6, 6, 5, and 5, with an average score of 5.5.

The reviewers raised some important concerns in their reviews and the authors responded with explanations and additional experiments. The details are as follows.

- (C1) Reviewers 639N and VhEv pointed out that the motivation of this paper was unclear. In response, the authors provided explanations about the motivation.
- (C2) Reviewer VhEv suggested more rigorous ablation studies on the components of the proposed method. In response, the authors supplemented experiments on the feature classifier and stated that the initial submission had contained ablations on other components.
- (C3) Reviewers PYLd, sywP, and VhEv suggested more evaluations on biases beyond the length bias. In response, the authors explained that they had evaluated other biases with the KL divergence between feature distributions.
- (C4) Reviewer PYLd was concerned that the performance improvement of PFP over the baselines is insignificant. In response, the authors conducted additional experiments with different random seeds.
- (C5) Reviewers sywP and VhEv suggested more baselines. In response, the authors added SimPO [1] as a new baseline, while refusing to compare PFP with OPRO [2] and RLCD [3] mentioned by Reviewer VhEv.

During the discussion period, Reviewer 639N raised the score because most of the concerns were addressed, and Reviewer VhEv (score: 5) still had Concerns C2, C3, and C5. Reviewers sywP (score: 6) and PYLd (score: 5) did not respond to the authors' rebuttal. None of the reviewers participated in the AC-Reviewers discussion.

I have carefully read the paper, the reviews, the authors' rebuttal, and the discussions between the authors and the reviewers. In my final decision, Concerns C2, C3, and C5 weigh most heavily, as the authors have not adequately addressed them.



[1] SimPO: Simple Preference Optimization with a Reference-Free Reward

[2] Large Language Models as Optimizers.

[3] RLCD: Reinforcement Learning from Contrastive Distillation for LM Alignment.

---

### Decision · Program_Chairs · 2025-01-22

Reject